# SAMHD1 deacetylation by SIRT1 promotes DNA end resection by facilitating DNA binding at double-strand breaks

Priya Kapoor-Vazirani[1], Sandip K. Rath[1], Xu Liu[2], Zhen Shu[1], Nicole E. Bowen [3], Yitong Chen[1], Ramona Haji-Seyed-Javadi[1], Waaqo Daddacha[4], Elizabeth V. Minten[1], Diana Danelia [1], Daniela Farchi[1], Duc M. Duong[2], Nicholas T. Seyfried [2], Xingming Deng[1], Eric A. Ortlund [2], Baek Kim [3] & David S. Yu [1] ✉

Sterile alpha motif and HD domain-containing protein 1 (SAMHD1) has a dNTPase-independent function in promoting DNA end resection to facilitate DNA double-strand break (DSB) repair by homologous recombination (HR); however, it is not known if upstream signaling events govern this activity. Here, we show that SAMHD1 is deacetylated by the SIRT1 sirtuin deacetylase, facilitating its binding with ssDNA at DSBs, to promote DNA end resection and HR. SIRT1 complexes with and deacetylates SAMHD1 at conserved lysine 354 (K354) specifically in response to DSBs. K354 deacetylation by SIRT1 promotes DNA end resection and HR but not SAMHD1 tetramerization or dNTPase activity. Mechanistically, K354 deacetylation by SIRT1 promotes SAMHD1 recruitment to DSBs and binding to ssDNA at DSBs, which in turn facilitates CtIP ssDNA binding, leading to promotion of genome integrity. These findings define a mechanism governing the dNTPase-independent resection function of SAMHD1 by SIRT1 deacetylation in promoting HR and genome stability.

DNA double-strand breaks (DSB) are highly cytotoxic lesions, which must be repaired in order to preserve genome integrity. DSBs are induced by multiple sources, including ionizing radiation (IR), chemical exposures, reactive oxygen species (ROS), collapse of stalled replication forks, and immunoglobulin variable (diversity) joining [V(D)J] and class switch recombination (CSR). Failure to repair DSBs can cause cell death or mutagenic events that lead to carcinogenesis, premature aging, neurodegeneration, infertility, and developmental and immunological abnormalities[1,2]. Homologous recombination (HR) is the most accurate DSB repair pathway and relies on a homologous sequence in a sister chromatid as a repair template, which restricts it to S and G2 phases of the cell cycle[3]. HR is initiated by the resection of DSB ends, which involves 5′ to 3′ nucleolytic degradation resulting in a long 3′ single-stranded DNA (ssDNA) tail[4]. The endo/exonuclease MRN complex, comprised of MRE11, RAD50, and NBS1, collaborates with the C-terminal-binding protein interacting protein (CtIP) endonuclease for initial short-range DNA resection[5–12]. Long-range extension of DNA resection is subsequently mediated by EXO1 and DNA2 together with WRN and BLM[13–18]. Resected ssDNA is rapidly coated by replication protein A (RPA) to prevent nucleolytic degradation, which is then displaced by RAD51[19]. The RAD51-DNA nucleoprotein filament then invades a homologous sequence in a sister chromatid and catalyzes ATP-dependent DNA strand exchange.

Sterile alpha motif (SAM) and histidine-aspartic acid (HD) domain-containing protein 1 (SAMHD1) is a deoxyribonucleoside (dNTP) triphosphohydrolase[20,21] with a well-defined role in restricting human

[1]Department of Radiation Oncology and Winship Cancer Institute, Emory University School of Medicine, Atlanta, GA 30322, USA. [2]Department of Biochemistry, Emory University School of Medicine, Atlanta, GA 30322, USA. [3]Department of Pediatrics, Emory University School of Medicine, Atlanta, GA 30322, USA. [4]Department of Biochemistry and Molecular Biology, Medical College of Georgia at Augusta University, Augusta, GA 30912, USA. ✉e-mail: dsyu@emory.edu

immunodeficiency virus-1 (HIV-1) and other viral infections in non-dividing immune cells[22–32] that has been attributed to its activity in depleting cellular dNTPs required for reverse transcription[20,28,31,33,34]. Mutations in *SAMHD1* are associated with Aicardi-Goutières syndrome[35], an inherited autoimmune encephalopathic disorder, and *SAMHD1* is mutated or downregulated in a number of cancers[36], including leukemias[37–39], lymphomas[40–42], colorectal cancer[43], and lung cancer[44], suggesting a role for SAMHD1 as a tumor suppressor. SAMHD1 contains an amino-terminal SAM domain[45], a protein inter-action domain[46], and a central HD domain[47], which contains its catalytic core. Crystal structure and biochemical studies show that SAMHD1 is allosterically activated by binding of dGTP/GTP and dNTP to two adjacent allosteric binding sites[48–53], which induces oligomerization of two inactive dimers to a dNTPase active tetramer[49,51,53,54]. In addition to its catalytic dNTPase activity, SAMHD1 binds to single-strand DNA/RNA (ssDNA/RNA)[32,55–62] at its dimer-dimer interface[59,62], which facilitates its dimerization[60–62] but sterically blocks tetramerization[59] required for its dNTPase activity[49,50,53,54,63], and binding of SAMHD1 to phos-phorthioated oligonucleotides at its allosteric sites promotes forma-tion of a mixed occupancy tetramer that contributes to an anti-viral activity that is suggested to be independent of its dNTPase activity[62]. Moreover, phosphorylation of SAMHD1 at threonine 592 (T592) inhi-bits its anti-viral activity but not dNTPase activity[32,64–67] while sumoy-lation of SAMHD1 at lysine 595 (K595) is required for its anti-viral but not dNTPase activity[68], providing further support for a dNTPase-independent contribution to viral restriction.

SAMHD1 has a dNTPase-independent function in genome main-tenance by promoting DNA end resection to facilitate DSB repair by HR through the recruitment of CtIP to DSBs[69] and resection of nascent DNA at stalled replication forks by stimulating MRE11 exonuclease activity[56]. Consistent with its role in DNA repair and binding to nucleic acids, SAMHD1 localizes to DSBs[37,69,70], stalled replication forks[56], and DSBs at immunoglobulin class switch regions induced by activation-induced cytidine deaminase (AID)[70]. SAMHD1 also acts at DSBs to restrain aberrant nucleotide insertions during DNA end joining[70,71], including at immunoglobulin class switch regions during CSR and *IgH/c-Myc* translocation processes[70], that is dependent on its dNTPase activity[70,71]. Furthermore, SAMHD1 prevents the accumulation of R loops, nucleic acid structures composed of a DNA/RNA hybrid and a displaced non-template ssDNA, at transcription-replication conflict regions[72]. SAMHD1 phosphorylation at T592 has been reported to promote its fork resection activity through an undefined mechanism[56]; however, the mechanisms governing the dNTPase-independent resection function of SAMHD1 at DSBs, as well as its binding to nucleic acids and localization to DNA damage sites are unclear.

Sirtuins are NAD⁺ dependent deacetylases, which regulate multi-ple biological processes, including genome maintenance, aging, tumorigenesis, and metabolism[73–80]. There are 7 mammalian homologs of yeast Sir2 that exhibit distinct subcellular localization, with SIRT1, SIRT6, and SIRT7 localized to the nucleus, SIRT2 residing mostly in the cytoplasm but also found in the nucleus, and SIRT3-5 localized to the mitochondria. SIRT1 is the closest homolog to yeast Sir2 and the most widely studied mammalian sirtuin[81]. Significantly, mice deficient in *Sirt1* develop sarcomas, lymphomas, teratomas, and carcinomas in a p53-deficient heterozygous background[82,83], and *Sirt1* overexpression in mice improves healthy aging[84], protects from metabolic syndrome-associated cancer[84], and suppresses age-dependent transcriptional changes[82] that are attributed at least in part to its role in promoting genome integrity[82–84]. In this regard, SIRT1 deacetylates BRG1 to pro-mote HR[85], and WRN, a SIRT1 substrate[86], is required for SIRT1-mediated HR[87]. SIRT1 also deacetylates NBS1[88], although NBS1 is not required for SIRT1-mediated HR[87], and a clearly defined role for SIRT1 in promoting DNA end resection has not been established. Further-more, SAMHD1 is acetylated at K405 by arrest defective protein 1(ARD1), which promotes its dNTPase activity[89]; however, the role of

SIRT1 deacetylation, and more generally of sirtuin and histone deace-tylase (HDAC) deacetylation, in directing SAMHD1 function, as well as the upstream signaling events governing SAMHD1's dNTPase-independent resection function in DSB repair are not known.

Here, we show that the dNTPase-independent resection function of SAMHD1 is regulated by SIRT1 deacetylation. SAMHD1 deacetylation by SIRT1 at conserved K354 in response to DSBs promotes DNA end resection and HR by facilitating its recruitment to DSBs and direct binding to ssDNA at DSBs, which in turn facilitates CtIP ssDNA binding to promote genome integrity. Our findings define a mechanism gov-erning the dNTPase-independent function of SAMHD1 in promoting DNA end resection and HR by SIRT1 deacetylation, elucidating a critical upstream signaling event governing the dNTPase-independent resec-tion function of SAMHD1 in promoting HR and genome stability and delineating a role for SIRT1 in promoting DNA end resection through SAMHD1 deacetylation.

## Results
### SIRT1 interacts with and deacetylates SAMHD1 in response to DSBs
To determine if SAMHD1 is regulated by acetylation in response to DSBs, we examined for acetylation of endogenous SAMHD1 in response to ionizing radiation (IR) and camptothecin (CPT), which induce DSBs, in several human cell lines. Western blot of immunoprecipitated (IP'ed) endogenous SAMHD1 with an anti-acetyl lysine antibody showed that endogenous SAMHD1 is acetylated at baseline in 293 T, HeLa, HCT116, and U2OS cells and deacetylated in response to IR and CPT (Fig. 1a). In contrast, endogenous acetylated SAMHD1 was not deacetylated in response to ultraviolet light (UV), a crosslinking agent, and methyl methanesulfonate (MMS), an alkylating agent (Supplementary Fig. 1a). These findings suggest that endogenous SAMHD1 is an acetylated pro-tein at baseline, deacetylated specifically in response to DSBs, and that this phenotype is generalizable to multiple cell types.

To determine if the DSB-regulated deacetylation of SAMHD1 is regulated by a sirtuin or non-sirtuin HDAC, IR-treated 293 T cells were pre-incubated with trichostatin A (TSA), which inhibits Class I, II, and IV HDACs, or nicotimamide (NAM), which inhibits Class III HDACs, also known as sirtuins. NAM but not TSA rescued the IR-regulated deacety-lation of endogenous SAMHD1 (Fig. 1b), suggesting that the DSB-regulated deacetylation of SAMHD1 is directed by a sirtuin. To provide insight into the sirtuin that deacetylates SAMHD1, we co-IP'ed SAMHD1-GFP expressed in 293T cells treated with and without IR and analyzed for pull down of the non-mitochondrial sirtuins SIRT1, SIRT2, SIRT6 and SIRT7 that localize to the nucleus. SAMHD1-GFP co-IP'ed with endo-genous SIRT1 but not endogenous SIRT2, SIRT6, or SIRT7 in response to IR (Fig. 1c), and furthermore SIRT1-FLAG expressed in 293T cells co-IP'ed with endogenous SAMHD1 in response to IR (Fig. 1d). Moreover, endo-genous SAMHD1 and SIRT1 co-IP'ed with each other in response to IR (Fig. 1e) validating their physiological interaction in a damage-regulated manner. The interaction between SAMHD1 and SIRT1 was direct as determined by in-vitro GST pulldown experiments with bacterially-purified recombinant SAMHD1 and GST-SIRT1 (lacking N-terminal amino acids 1–192; Supplementary Fig. 1b, c). To determine if the IR-regulated deacetylation of SAMHD1 is dependent on SIRT1 or other nuclear sir-tuins, we silenced SIRT1, SIRT2, SIRT6 and SIRT7 in 293 T cells treated with and without IR. The IR-regulated deacetylation of endogenous SAMHD1 was alleviated by depletion of SIRT1 but not SIRT2, SIRT6, or SIRT7 (Fig. 1f, g). Moreover, inhibition of SIRT1 deacetylase activity with Ex-527[90] alleviated the IR-regulated deacetylation of endogenous SAMHD1 (Fig. 1g), implying that SAMHD1 deacetylation in response to IR is dependent on SIRT1 deacetylase activity. Finally, overexpression of SIRT1-FLAG wild-type (WT) but not catalytically inactive H363Y[91] dea-cetylated endogenous SAMHD1 expressed in 293T cells, and deacety-lation was impaired by Ex-527 (Fig. 1h). Collectively, these findings strongly indicate that endogenous SAMHD1 directly complexes with

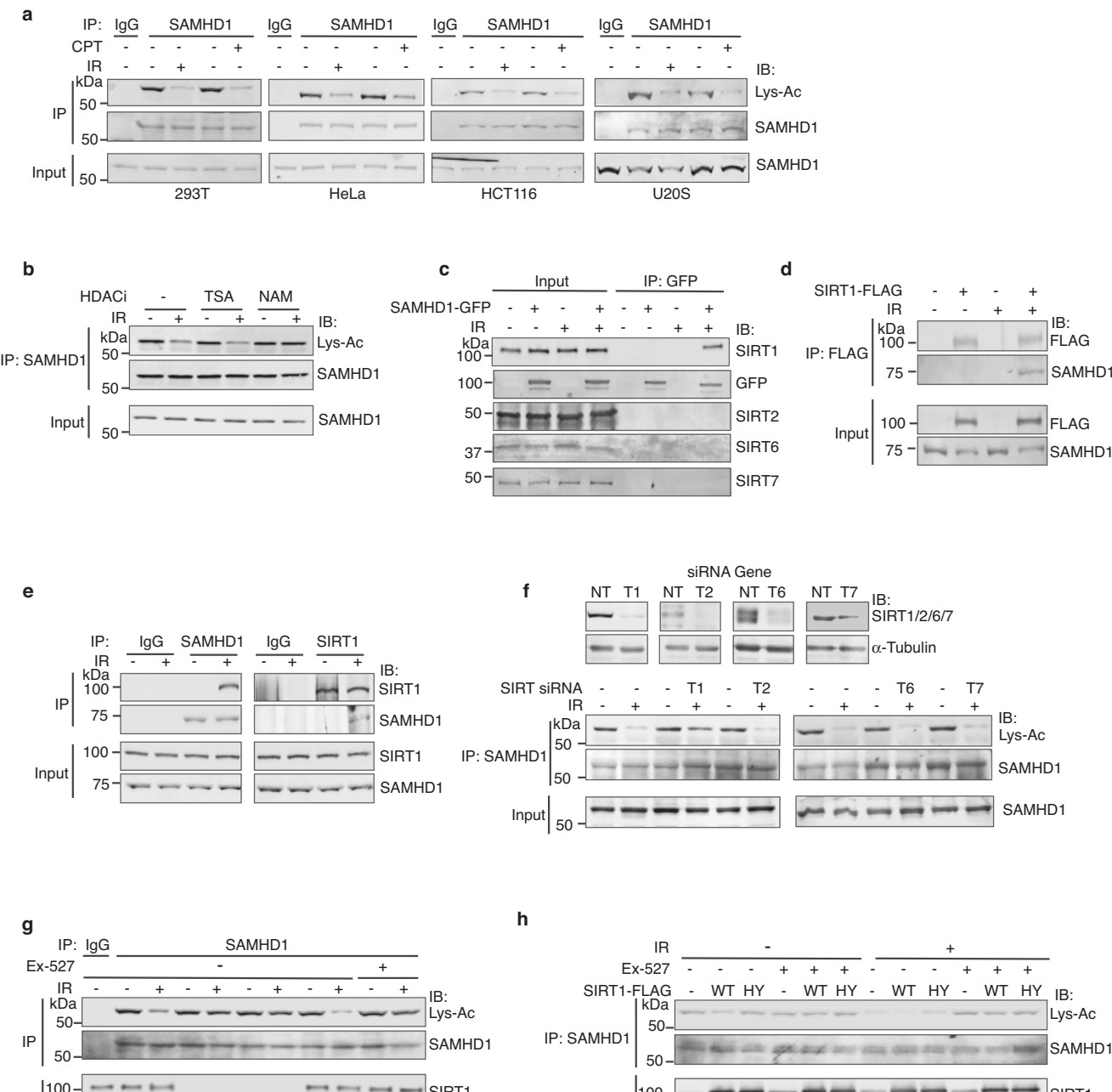

**Fig. 1 | SIRT1 interacts with and deacetylates SAMHD1 in response to DSBs.**
**a** Indicated cell lines were treated with or without 5 μM CPT or 10 Gy IR and harvested after 4 h (hrs). Protein lysates were immunoprecipitated (IP'ed) with IgG or anti-SAMHD1 antibody. Input and IP'ed proteins were run on SDS-PAGE gel and western blotted with indicated antibodies. **b** 293T cells were pre-treated with either 0.5 μM TSA or 10 mM NAM for 1 h and then additionally treated for 4 h in the presence of 10 Gy IR. Protein lysates were IP'ed with anti-SAMHD1 antibody. Input and IP'ed lysates were separated by SDS-PAGE and immunoblotted with indicated antibodies. **c**, **d** 293T cells or 293T cells overexpressing SAMHD1-GFP (**c**) or SIRT1-FLAG (**d**) were treated with or without 10 Gy IR and harvested after 4 h. Protein lysates processed from the cells were IP'ed with the indicated antibody. Input and IP'ed proteins were run on SDS-PAGE and immunoblotted with indicated antibodies. **e** 293T protein lysates treated with or without IR were IP'ed with the indicated antibodies, run on SDS-PAGE, and immunoblotted with anti-SIRT1 and

SAMHD1 antibodies. **f** Expression level of the indicated sirtuin proteins in 293T cells transfected with non-targeting siRNA (NT) or siRNA against the indicated sirtuin (SIRT1, T1; SIRT2; T2; SIRT6; T6; SIRT7; T7) was determined by western blot. Three days post-transfection, cells were incubated for 4 h with or without 10 Gy IR and processed for IP with anti-SAMHD1 antibody. Input and pull-down proteins were separated by SDS-PAGE and immunoblotted with indicated antibodies (bottom). **g** 293T cells were silenced for SIRT1 or a NT control or treated with the SIRT1 inhibitor Ex-527 (1 μM) where indicated for 1 h, treated with or without 10 Gy IR, and harvested 4 h later. Protein lysates were IP'ed with anti-SAMHD1 antibody. Input and IP'ed proteins were run on SDS-PAGE and immunoblotted with indicated antibodies. **h** 293T cells over-expressing wild-type (WT) or catalytically-dead (HY) SIRT1 were treated with Ex-527 and IR and processed as described in **g**. Source data are provided as a Source Data file.

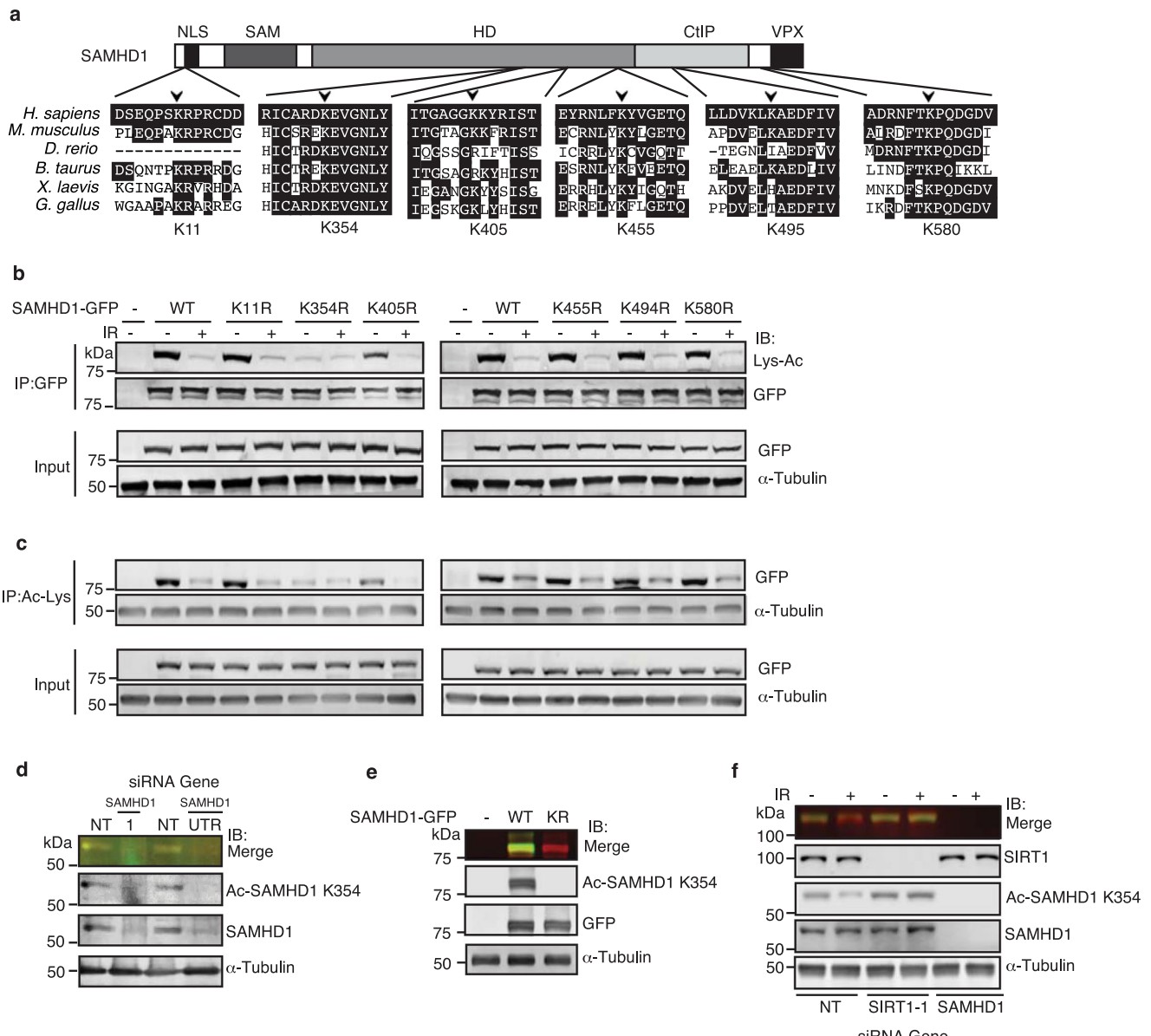

**Fig. 2 | SIRT1 deacetylates SAMHD1 at K354 in response to DSBs. a** Schematic of SAMHD1 protein depicting key domains and lysine residues tested in **b** and **c** for acetylation are shown. NLS nuclear localization signal, SAM sterile alpha-motif protein interaction domain, HD histidine-aspartate catalytic domain, CtIP CtIP-interacting domain, Vpx Viral Vpx-binding domain. SAMHD1 sequence alignment across species showing evolutionary conservation of lysine residues tested for acetylation is also shown. **b, c** SAMHD1-GFP WT or non-acetylable lysine to arginine (KR) mutants were overexpressed in 293 T cells and protein lysates from cells (±10 Gy IR) were IP'ed with anti-GFP (in **b**) or acetylated lysine (Lys-Ac; in **c**) antibodies. Input and IP'ed products were separated by SDS-PAGE and blotted with

indicated antibodies. **d, e** Validation of custom site-specific anti-acetyl SAMHD1 K354 antibody. **d** Protein lysates from 293 T cells silenced for SAMHD1 using two different siRNAs (1 and UTR) or a NT control were western blotted with antibodies that recognize SAMHD1 or SAMHD1 specifically acetylated at lysine 354 (Ac-SAMHD1 K354). **e** Protein lysates from 293T and 293T cells over-expressing SAMHD1-GFP WT or SAMHD1-GFP K354R (KR) were probed with indicated antibodies. **f** 293T and 293T cells silenced for SIRT1 or SAMHD1 were exposed to IR (10 Gy for 4 h) and processed for western blot with indicated antibodies. Source data are provided as a Source Data file.

and is deacetylated by SIRT1 but not other nuclear sirtuins in response to DSBs.

## SIRT1 deacetylates SAMHD1 at K354 in response to DSBs

To identify the specific lysine (K) target sites of SIRT1 deacetylation on SAMHD1, we performed mass spectrometry of purified SAMHD1-HA expressed in 293T cells treated with and without IR or CPT and identified candidate acetylation sites at K354, K494, and K580 with enrichment of K354 acetylation before but not after IR and CPT and K494 and K580 acetylation only after IR (Fig. 2a and Supplementary Fig. 2a, b). Several additional candidate acetylation sites on SAMHD1 have been identified in other high throughput mass spectrometry

analyses[89,92], including K11, K405, K455, and K580 with validation of K405 but not K580[89]. Amongst these sites, K354 with its surrounding amino acids is the most evolutionarily conserved (Fig. 2a), indicating its potential functional significance. To determine which lysine sites on SAMHD1 are acetylated and subsequently deacetylated in response to IR, we generated SAMHD1-GFP mutants, where a single lysine was replaced by arginine, which mimics non-acetylable lysine, and expressed the mutants in 293 T cells followed by treatment with and without IR. In contrast to SAMHD1 K11R, K455R, K494R, and K580R, which showed comparable acetylation to SAMHD1 WT at baseline, SAMHD1 K354R and to a lesser extent K405R were acetylated at lower levels, as determined by IP of SAMHD1-GFP and western blot with an

anti-acetyl-lysine antibody and by IP of acetyl-lysine proteins and western blot with GFP antibody (Fig. 2b, c), suggesting that both K354 and K405 contribute to SAMHD1 acetylation. Furthermore, IR induced a decrease in acetylation in all of the mutants except K354R, suggesting that K354 is the major site of deacetylation in response to IR. To validate acetylation of endogenous SAMHD1 at K354, we generated and validated a rabbit polyclonal anti-acetyl SAMHD1 K354 antibody. This antibody recognizes endogenous SAMHD1 and exogenous SAMHD1-GFP WT but not K354R expressed in 293T cells where the bands are also recognized by an anti-SAMHD1 antibody and can be silenced with SAMHD1 siRNA (Fig. 2d, e). To determine whether endogenous SAMHD1 is deacetylated at K354 in response to IR by SIRT1, we performed rescue experiments in 293 T cells treated with and without IR and/or SIRT1 depletion using our anti-acetyl SAMHD1 K354 antibody. Endogenous SAMHD1 was deacetylated at K354 in response to IR, and this was rescued by SIRT1 depletion (Fig. 2f), strongly suggesting that SIRT1 deacetylates SAMHD1 at K354 in response to DSBs.

## SAMHD1 deacetylation at K354 by SIRT1 promotes HR and DNA end resection

Given that SAMHD1 has a dNTPase independent function in promoting DNA end resection to facilitate HR of DSBs induced by IR[69] and that K354 is deacetylated in response to IR, we investigated whether SAMHD1 deacetylation at K354 might be important for HR and DNA end resection. To determine directly if SAMHD1 deacetylation at K354 promotes HR, we performed rescue experiments in U2OS cells integrated with a direct repeat (DR)-GFP HR reporter substrate in which expression of the I-SceI rare cutting endonuclease generates a DSB that when repaired by HR restores GFP expression[93]. SAMHD1 depletion impaired HR, which was rescued by expression of SAMHD1-RFP WT and K354R but not acetyl mimic K354Q where lysine is replaced by glutamine (Q) (Fig. 3a, b and Supplementary Fig. 3a, c), suggesting that SAMHD1 deacetylation at K354 promotes HR. Consistent with a role for K354 deacetylation in promoting HR, SAMHD1 depletion caused hypersensitivity to velipirab, a PARP inhibitor, which was rescued by expression of SAMHD1-GFP WT and K354R but not K354Q (Supplementary Fig. 4). To determine if SAMHD1 deacetylation at K354 promotes DNA end resection, we examined for IR-induced phosphorylation of RPA32 at serine 4 and 8 (S4/8), a marker of DNA end resection, in 293T cells silenced for SAMHD1 and expressing SAMHD-GFP WT, K354R, and K354Q. SAMHD1 depletion impaired IR-induced phosphorylation of RPA32 at S4/8, and this was rescued by expression of SAMHD1-GFP WT and K354R but not K354Q (Fig. 3c), suggesting that SAMHD1 deacetylation at K354 promotes DNA end resection. To more directly determine if K354 deacetylation promotes DNA end resection, we labeled U2OS cells with BrdU, treated the cells with IR, and probed the cells for BrdU under nondenaturing conditions, which labels ssDNA. SAMHD1 depletion also impaired BrdU but not γH2AX foci under these conditions, which was rescued by expression of SAMHD1-GFP WT and K354R but not K354Q (Fig. 3d, e). To determine if SIRT1 and SAMHD1 function together in promoting HR, we performed an epistasis study with the DR-GFP assay. Both SIRT1 and SAMHD1 depletion in cells impaired HR, and combined depletion of SIRT1 and SAMHD1 caused no further impairment in HR (Fig. 3f, g and Supplementary Fig. 3b, d), suggesting that SIRT1 and SAMHD1 function together in promoting HR. To determine if K354 deacetylation by SIRT1 promotes HR, we performed rescue experiments following SIRT1 depletion with expression of SAMHD1-RFP WT, K354R, and K354Q. Expression of SAMHD1-RFP WT and K354R but not K354Q alleviated the HR impairment of SIRT1 depletion (Fig. 3h, i and Supplementary Fig. 3a, e), suggesting that SIRT1 promotes HR at least in part through SAMHD1

K354 deacetylation. Similarly, expression of SAMHD1-GFP K354R but not K354Q rescued the impairment in IR-induced RPA32 S4/8 phosphorylation of SIRT1 depletion (Fig. 3j), suggesting that K354 deacetylation functions downstream of SIRT1 in promoting DNA end resection. Collectively, these data indicate that SAMHD1 deacetylation at K354 by SIRT1 promotes HR and DNA end resection.

## SAMHD1 K354 deacetylation does not direct its tetramerization and dNTPase activities

SAMHD1 is known to form a tetramer in the presence of dGTP/GTP and dNTP[49–51,53,54,94], and tetramerization has been reported to be required for its dNTPase and HIV-1 restriction activities[49,50,53,54,63]. Since K354 resides around the second allosteric binding site of dGTP, based on previously solved crystal structures of the human tetrameric SAMHD1-dGTP complex[50,51,53], we sought to determine whether SAMHD1 K354 deacetylation regulates SAMHD1 tetramerization. SAMHD1 WT, K354R and K354Q mutant proteins were analyzed for their oligomerization status in the presence or absence of dGTP using size exclusion chromatography (SEC). Purified SAMHD1 with an N-terminal His tag and linker region has a calculated molecular weight of 74 kDa (Fig. 4a). SAMHD1 WT protein eluted at 14.8 mL, between the molecular standards of 158 kDa (13.5 mL) and 44 kDa (16 mL) (Fig. 4b). Preincubation with excess dGTP caused SAMHD1 to predominantly elute at 12.0 mL, between the molecular standards of 670 kDa (11.3 mL) and 158 kDa (13.5 mL), suggesting the formation of a dGTP-induced tetramer (Fig. 4c). Likewise, preincubation with excess dGTP induced tetramerization of both SAMHD1 K354Q and K354R (Fig. 4d, e), suggesting that the acetylation status of K354 does not direct SAMHD1 oligomerization.

To determine if SAMHD1 K354 deacetylation directs its dNTPase activity, total cellular dNTPs were extracted from 293 T cells overexpressing SAMHD1-GFP WT, K354R, K354Q, and H206A/D207A (HD/AA), which disrupts SAMHD1's active site and impairs its dNTPase activity[20,30] (Fig. 4f). A RT-based primer extension assay to quantify dNTP pools[33] showed that overexpression of SAMHD1-GFP WT, K354R, and K354Q but not HD/AA all lead to a comparable decrease in total dATP, dCTP, dGTP, and dTTP pools (Fig. 4g), suggesting that SAMHD1 deacetylation at K354 does not direct its dNTPase activity.

## SIRT1 deacetylation of K354 facilitates SAMHD1 localization to DSBs and SAMHD1 residues 300–465 is sufficient for this localization

To determine if SAMHD1 deacetylation by SIRT1 is important for its localization to DSBs, we first examined U2OS cells treated with IR following SIRT1 depletion or SIRT1 inhibition with Ex-527. Both SIRT1 depletion and SIRT1 inhibition impaired the co-localization of endogenous SAMHD1 with IR-induced DSBs marked by γH2AX (Fig. 5a and Supplementary Fig. 5a). Moreover, SIRT1 depletion and SIRT1 inhibition impaired the co-localization of SAMHD1 with mCherry-LacI-FokI endonuclease-induced DSBs in U2OS reporter cells integrated with lac operator repeats[95] (Fig. 5b and Supplementary Fig. 5b), suggesting that SIRT1 deacetylase activity promotes SAMHD1 localization to DSBs. To identify the region of SAMHD1 that mediates its recruitment to DSBs, we generated SAMHD1-GFP deletion mutants (Fig. 5c, d) and examined their co-localization with FokI-induced DSBs in U2OS cells. Mutants lacking SAMHD1's nuclear localization signal (NLS), KRPR at amino acids 11–14[63,96], were fused to myc-NLS to ensure nuclear localization. In contrast to SAMHD1-GFP (1–115), which contains the SAM domain, and (115–300), which contains the amino-terminal core catalytic HD domain, SAMHD1-GFP (300–465), which contains the carboxyl-terminal HD domain but not SAM, CtIP binding, and Vpx-binding domains, co-localized with Fok-I induced DSBs (Fig. 5e, f), suggesting that SAMHD1 (300–465) is sufficient for localization to DSBs. Similarly, SAMHD1-GFP

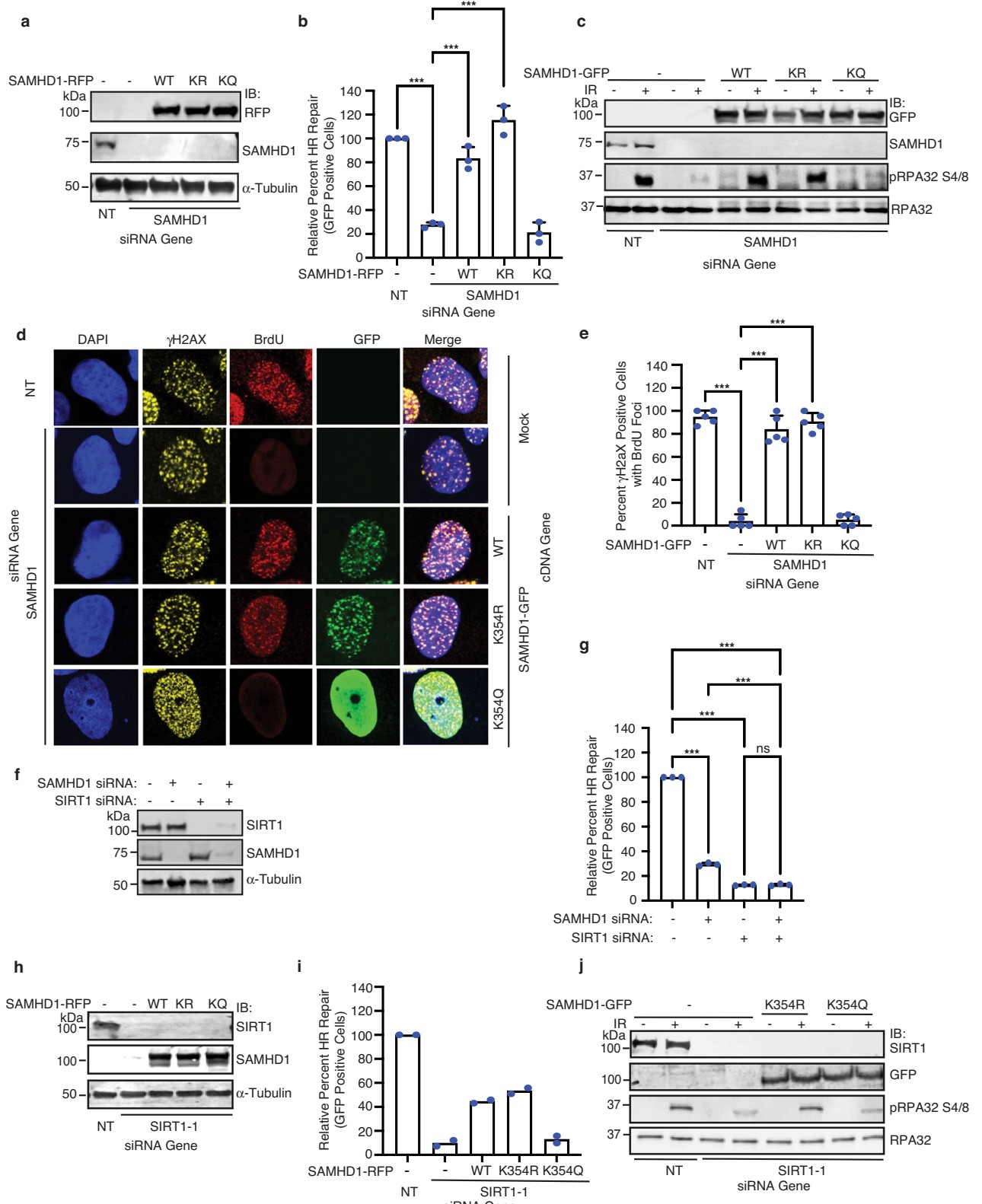

(300–465) but not SAMHD1-GFP (1–115) or (115–300) co-IP'ed with SIRT1-FLAG in irradiated cells (Supplementary Fig. 5c). Furthermore, SAMHD1-GFP K354R but not K354Q co-localized with FokI-induced DSBs (Fig. 5e, f), suggesting that SAMHD1 deacetylation at K354 promotes its localization to DSBs. Collectively, these data imply that SIRT1 deacetylation of K354 within a DNA damage recruitment domain-containing amino acids 300–465 facilitates SAMHD1 recruitment to DSBs.

## SAMHD1 K354 deacetylation promotes binding with ssDNA and ssDNA overhangs and AsiSI-induced DSBs

SAMHD1 complexes with and co-localizes with both the CtIP and MRE11 DNA end resection proteins in response to DSBs[56,69]. To determine whether SAMHD1 deacetylation at K354 may direct its interactions with these proteins, we performed co-IP of SAMHD1-GFP WT, K354R, and K354Q expressed in 293T cells treated with or without IR

**Fig. 3 | SAMHD1 deacetylation at K354 by SIRT1 promotes HR and DNA end resection. a** Western blot showing SAMHD1-RFP expression and SAMHD1 down-regulation in the DR-GFP assay from **b**. **b** U2OS cells containing an integrated DR-GFP HR reporter were silenced for SAMHD1 and transfected with the I-SceI endo-nuclease and RFP (−) or SAMHD1-RFP. Live cells were subjected to flow cytometry. Percent GFP positive cells within the RFP-expressing population was determined for HR efficiency. Mean and standard deviation (SD) from three independent replicas is shown. **c** 293T cells with or without SAMHD1 and expressing SAMHD1-GFP were irradiated and processed for western blot with the indicated antibodies. **d, e** U2OS cells transfected with SAMHD1 siRNA and SAMHD1-GFP constructs were incubated with 30 μM BrdU for 36 h and then treated with 10 Gy IR for 4 h. Cells were fixed under non-denaturing conditions and processed for immuno-fluorescence microscopy with anti-γH2AX and BrdU antibodies. DNA was stained with DAPI and GFP fluorescence was determined. Shown is a representative image (**d**) and quantification indicating relative percent γH2AX positive cells with BrdU

foci (**e**). Quantification represents mean and SD from five independent replicas. **f** Western blot showing SAMHD1 and/or SIRT1 silencing in cells used in **g**. **g** U2OS DR-GFP reporter cells expressing I-SceI endonuclease were silenced for SAMHD1, SIRT1 or both. Live cells were subjected to flow cytometry and sorted for GFP positive cells. Shown are the mean and SD from three independent experiments. **h** Western blot depicting SAMHD1-RFP protein expression and SIRT1 silencing in U2OS DR-GFP reporter cells used in **i**. **i** U2OS DR-GFP reporter cells silenced for SIRT1 and over-expressing SAMHD1-RFP proteins (where specified) were treated and processed as in **b**. Quantification represents mean from two independent replicas. **j** 293T and SIRT1-silenced 293T cells were transfected with SAMHD1-GFP constructs, where shown. Cells were irradiated, harvested and protein lysates were western blotted with indicated antibodies. For graphs in panels **b**, **e**, and **g**, $p$ values (***$p < 0.001$) were determined using Ordinary one-way ANOVA with Dunnett's post hoc test analysis. Source data are provided as a Source Data file.

but observed comparable pulldown of CtIP and endogenous MRE11, RAD50, and NBS1 (Fig. 6a), suggesting that K354 deacetylation is not critical for mediating interaction with CtIP or the MRN complex.

SAMHD1 binds to ssDNA but not double-strand DNA (dsDNA)[32,55–62]. Using streptavidin pulldown of biotin-labeled DNA structures incubated with 293T lysates, we found similarly no evidence that endogenous SAMHD1 binds to dsDNA with blunt ends but that endogenous SAMHD1 has a preference for binding to dsDNA struc-tures with 5′ ssDNA overhangs > ssDNA > dsDNA with 3′ ssDNA over-hangs and that these interactions are increased after IR (Fig. 6b and Supplementary Fig. 6a, b). These interactions were not disrupted by treatment with RNaseH1, which degrades RNA in DNA/RNA hybrids nor RNaseA, which degrades RNA (Supplementary Fig. 6c, d), consistent with a prior report that despite resolving R loops[72], SAMHD1 does not bind DNA/RNA hybrids[61] and suggesting that these interactions are not RNA-mediated. Recent studies using chemical crosslinking of ssRNA and ssDNA oligonucleotides with SAMHD1 and x-ray crystallography of synthetic phosphorothioated oligonucleotides with SAMHD1 suggest that K354 is located in a region critical for SAMHD1's interface with nucleic acids[59,62]. Since ssDNA and 5′ ssDNA overhangs bound SAMHD1 most efficiently, we examined the interaction of SAMHD-GFP WT and mutants with these biotin-labeled DNA structures. Both SAMHD1-GFP WT and K354R showed increased binding with ssDNA and 5′ssDNA overhangs compared with SAMHD1-GFP K354Q, and the interaction of SAMHD1-GFP WT with both of these structures was increased in response to IR (Fig. 6c), suggesting that K354 deacetylation in response to IR increases binding to ssDNA and 5′ssDNA overhangs. SAMHD1-GFP (300–465), which contains K354 and is sufficient to localize to Fok1-induced DSB sites, also showed increased binding with ssDNA in response to IR (Supplementary Fig. 6e). The increase in binding of endogenous SAMHD1 with ssDNA in response to IR was associated with an increase in binding of endogenous CtIP to ssDNA, and both the IR-induced increase in binding of SAMHD1 and CtIP with ssDNA was impaired with SIRT1 depletion (Fig. 6d). Moreover, SAMHD1 depletion impaired the IR-induced increase in binding of CtIP with ssDNA, which was rescued by expression of SAMHD1-GFP WT and K354R but not K354Q (Fig. 6e). Consistently, depletion of SIRT1 or SAMHD1 impaired CtIP localization to DSBs following IR (Supple-mentary Fig. 7). Collectively, these data suggest that K354 deacetyla-tion by SIRT1 in response to IR promotes SAMHD1 binding to ssDNA, which in turn facilitates CtIP localization to DSBs and binding to ssDNA.

To determine if K354 deacetylation by SIRT1 promotes SAMHD1 binding to endogenous DSBs, we performed chromatin IP (ChIP) with quantitative PCR (qPCR) using U2OS DIvA (DSB Inducible via AsiSI) cells, in which 4-hydroxytamoxifen (4OHT) treatment triggers nuclear localization of AsiSI to induce approximately 150 sequence-specific DSBs, dispersed across the genome[97]. SIRT1 depletion in DIvA cells impaired the enrichment of endogenous SAMHD1 to AsiSI-induced

DSBs (Fig. 6f, g). Furthermore, SAMHD1-GFP K354R was enriched to a significantly greater extent than SAMHD1-GFP K354Q to AsiSI-induced DSBs (Fig. 6h and Supplementary Fig. 6f). Together, these data imply that K354 deacetylation by SIRT1 facilitates SAMHD1 binding to DSBs.

To determine whether K354 deacetylation is critical for mediating a direct interaction with ssDNA, we performed an electrophoretic mobility shift assay (EMSA) with a chemically synthesized peptide of FLAG-SAMHD1 (332–384), which was acetylated or not acetylated at K354 (Supplementary Fig. 6g). A dose dependent shift in mobility of ssDNA was observed with an increasing amount of non-acetylated SAMHD1 (332–384) peptide but not appreciably with K354 acetylated SAMHD1 (332–384; Fig. 6i), suggesting that SAMHD1 (332–384) is sufficient for binding ssDNA and that K354 acetylation directly impairs SAMHD1 binding to ssDNA. In a complementary approach, streptavi-din pulldown of biotin-labeled ssDNA confirmed the binding of non-acetylated SAMHD1 (332–384) peptide to ssDNA, which was impaired by K354 acetylation (Supplementary Fig. 6h), implying that K354 deacetylation is critical for mediating a direct interaction with ssDNA.

## SAMHD1 K354 deacetylation promotes genome stability

To determine if K354 deacetylation is important for genome stability, we examined for the spontaneous accumulation of γH2AX and CHK2 phosphorylation at threonine 68 (p-CHK2 Thr68) foci, both markers of genomic instability. SAMHD1 depletion in U2OS cells caused an increase in γH2AX but not total H2AX levels, which was rescued by expression of SAMHD1-GFP WT and K354R but not K354Q (Fig. 7a). Similarly, SAMHD1 depletion in U2OS cells caused an increase in p-CHK2 Thr68 foci, which was rescued by expression of SAMHD1-GFP WT and K354R but not K354Q (Fig. 7b, c). Together, the data suggest that SAMHD1 deacetylation at K354 promotes genome stability.

## Discussion

Our findings reveal a regulatory mechanism governing the dNTPase-independent resection function of SAMHD1 whereby SAMHD1 deace-tylation at conserved K354 by SIRT1 facilitates SAMHD1 recruitment to DSBs and its direct binding to ssDNA at DSBs, which in turn facilitates CtIP ssDNA binding to promote DNA end resection and HR, leading to promotion of genome integrity. These findings elucidate a critical upstream signaling event directing SAMHD1 localization to DNA damage sites and binding to nucleic acids that promotes DNA end resection and HR, identify SAMHD1 as a unique interacting partner and substrate for SIRT1, and establish SIRT1 as a positive regulator of DNA end resection. Moreover, these findings help explain how *Sirt1* defi-ciency results in genomic instability and carcinogenesis. In this regard, we found that SIRT1 but not other nuclear sirtuins interacts with and deacetylates SAMHD1 at conserved K354, located in a region critical for interfacing with ssDNA, in response to IR but not UV or MMS, providing evidence that SAMHD1 is regulated specifically by SIRT1 in a DSB-regulated manner and that SAMHD1 is an interacting partner and

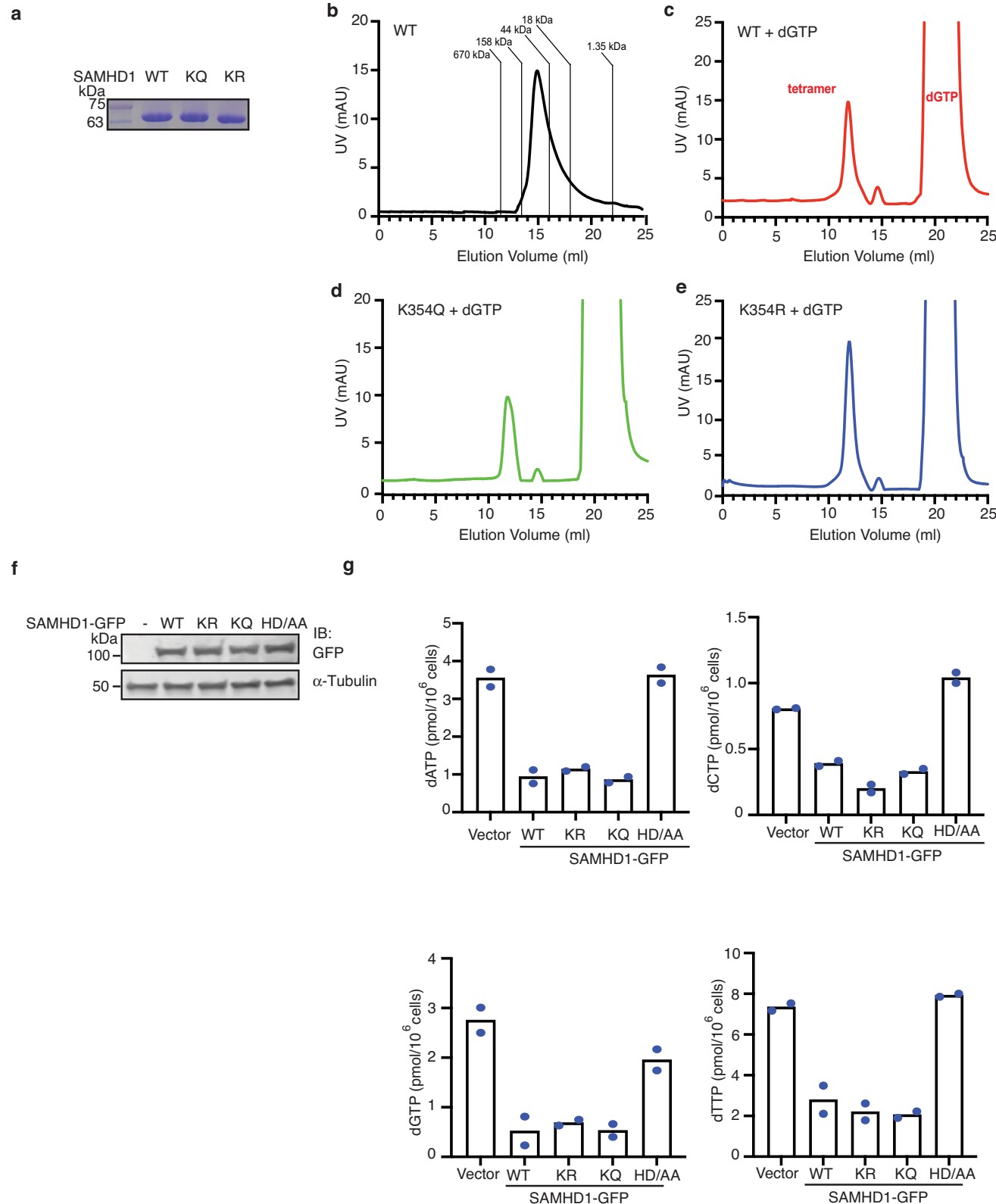

**Fig. 4 | SAMHD1 K354 deacetylation does not direct its tetramerization and dNTPase activities. a** Coomassie-stained gel showing the relative amounts of purified SAMHD1 proteins used in **b–e. b–e** SAMHD1 proteins expressed and purified from bacterial cells were incubated with or without 3 mM dGTP and subjected to size exclusion chromatography to determine its ability to tetramerize. Shown are the elution profiles with UV traces of excitation at 280 nm. **f** Western blot showing expression levels of SAMHD1-GFP constructs used in **g. g** Total cellular dNTPs were extracted from 293 T cells expressing SAMHD1 WT or mutants and subjected to RT-based primer extension assay to quantify total dATP, dCTP, dGTP, and dTTP concentrations. Shown are the average from two independent replicas. Source data are provided as a Source Data file.

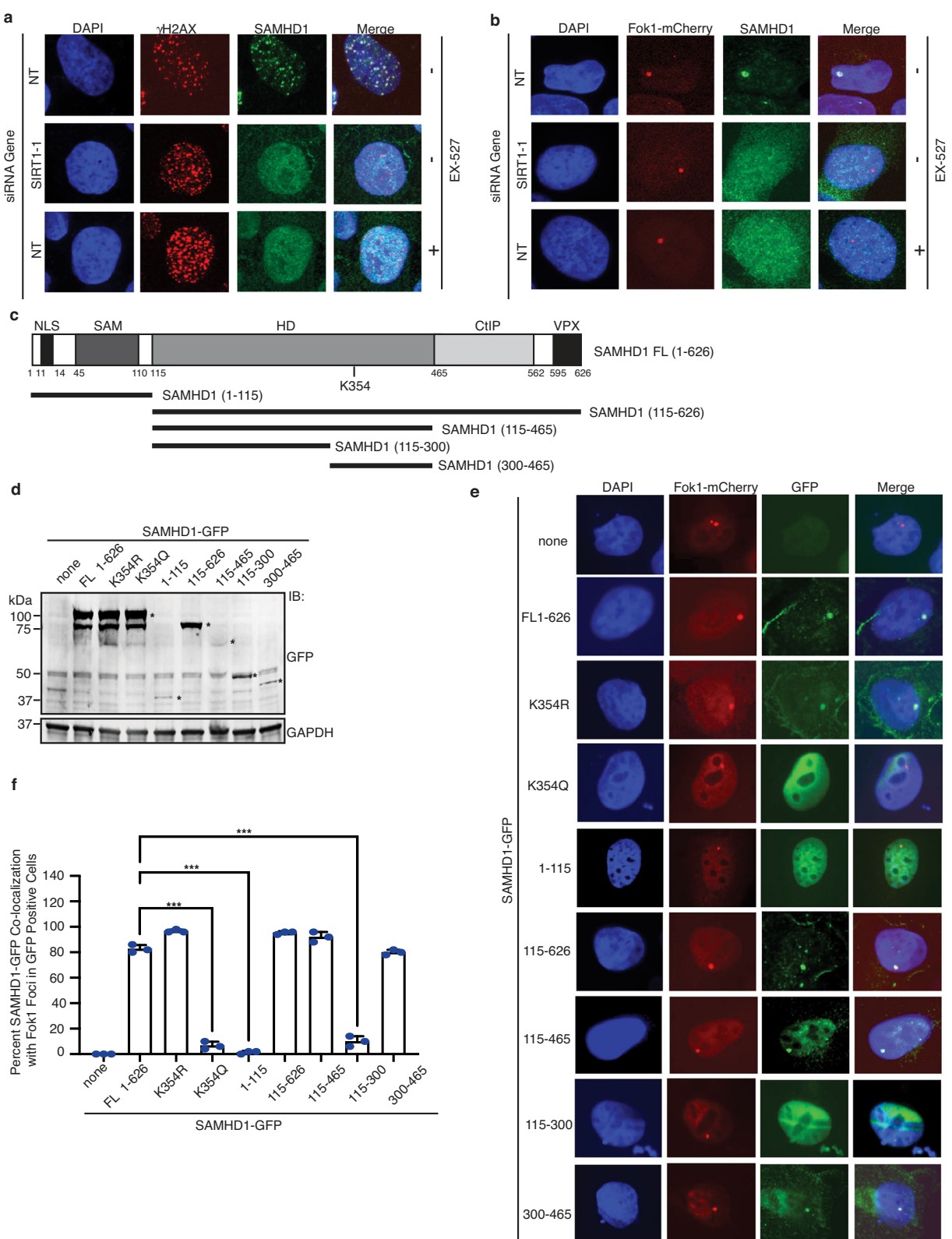

substrate for SIRT1. In addition, we found that K354 deacetylation by SIRT1 promotes DNA end resection and HR but not SAMHD1 tetramerization or dNTPase activity, demonstrating specificity and functional significance of K354 deacetylation by SIRT1 and establishing a role for SIRT1 in promoting DNA end resection. Furthermore, we found that K354 deacetylation by SIRT1 promotes SAMHD1 recruitment to DSBs and direct binding to ssDNA at DSBs, which in turn facilitates CtIP

ssDNA binding, and moreover that K354 deacetylation promotes genome integrity, providing a mechanistic model for how SAMHD1 deacetylation by SIRT1 promotes DNA end resection and HR and elucidating a regulatory mechanism for SAMHD1 localization to DNA damage sites and binding to nucleic acids. Thus, our findings support a model in which in response to DSBs, SIRT1 interacts with and deacetylates SAMHD1 at K354 to facilitate its recruitment to DSBs and direct

**Fig. 5 | SIRT1 deacetylation of K354 facilitates SAMHD1 localization to DSBs and SAMHD1 residues 300–465 is sufficient for this localization. a** U2OS, SIRT1-silenced U20S, and EX-527 treated-U2OS cells were exposed to IR (10 Gy, 4 h) and processed for immunofluorescence with indicated antibodies. Representative confocal microscopy images are shown. **b** U20S-265 Fok1 cells were treated with SIRT1 siRNA or EX-527 and induced for Fok1 expression with 1 μM Shield-1 and 2 μM 4-OHT for 4 h. Localization of Fok1 and SAMHD1 proteins were visualized by mCherry fluorescence and anti-SAMHD1 antibodies. DNA was visualized by DAPI staining. Representative images obtained by confocal microscopy are shown. **c** A schematic showing residues comprising full-length (FL) SAMHD1 and SAMDH1 truncation constructs used in **d**–**f**. Mutants lacking SAMHD1's nuclear localization

signal (NLS), KRPR at amino acids 11–14, were fused to myc-NLS to ensure nuclear localization. **d** Western blot depicting expression levels of the SAMHD1-GFP fragments used in **e**. Asterisk marks the size of the SAMHD1 fragments. **e**, **f** U20S-265 Fok1 cells were transfected with indicated SAMHD1-GFP fragments and induced for Fok1 expression. Localization of Fok1 and SAMHD1 was determined by mCherry and GFP fluorescence, respectively, using confocal microscopy. Shown are representative images (**e**) and quantification showing percentage of GFP-expressing cells with SAMHD1-GFP and Fok1 co-localization (**f**). **f** Mean and SD from three replicas of 50 cells is shown. *P* values (***p < 0.001) were determined using Ordinary one-way ANOVA with Dunnett's post hoc test analysis. Source data are provided as a Source Data file.

binding to ssDNA at DSBs, which in turn facilitates CtIP binding to DSBs to promote DNA end resection and HR. Dysregulation of this pathway impairs DNA end resection and HR, leading to genomic instability (Fig. 7).

SAMHD1 has pleiotropic functions in genome maintenance, viral restriction, innate immune response, restriction of long-interspersed 1 (LINE-1) retrotransposition, and post-transcriptional control that are both dependent and independent of its dNTPase activity, involve nucleic acid binding, and/or localization to DSBs, suggesting the need for tight regulation[98]. We have identified a critical upstream signaling event in the damage-regulated deacetylation of K354 by SIRT1 in governing the dNTPase-independent resection function of SAMHD1 in HR by facilitating its localization to DSBs and binding to ssDNA at DSBs. However, we found no evidence that K354 deacetylation directs SAMHD1's tetramerization or dNTPase activity, suggesting specificity of K354 deacetylation in directing SAMHD1's dNTPase-independent functions. Similarly, SAMHD1 phosphorylation at T592 and sumoylation at K595 direct anti-viral activities independent of its dNTPase activity[32,64–66,68,99]. SAMHD1 phosphorylation at T592 has also been reported to promote its fork resection activity[56]. Given that T592 is located outside of SAMHD1's DNA binding domain at amino acids 332–384 and more broadly its DNA damage localization domain at amino acids 300–465, which is sufficient for its localization to DSBs, T592 phosphorylation may regulate SAMHD1's resection function through an alternative but complementary mechanism. Indeed, it was recently reported that while T592 phosphorylation has a minimal effect on ssDNA binding, it promotes SAMHD1 carboxyl-terminal domain dynamics and tetramer dissociation to facilitate ssDNA binding access[100]. Similar to a previous report, we found that SAMHD1 is acetylated at K405[89]; however, in contrast to K354, the acetylation status of K405, which regulates its dNTPase activity[89], did not change after IR, suggesting that control of SAMHD1's dNTPase activity by K405 acetylation is not regulated by DSBs. While SAMHD1 also acts at DSBs to restrain aberrant nucleotide insertions during DNA end joining through its dNTPase activity[70,71], it is possible that this regulation could be a multi-step process whereby SAMHD1's initial recruitment and binding to DSBs is regulated by K354 deacetylation by SIRT1.

SAMHD1 binds to ssDNA/RNA but not dsDNA/RNA or DNA/RNA hybrids[32,55–62]. We found similar data, and our findings further extend these data by showing that endogenous SAMHD1 has a preference for binding to dsDNA structures with 5′ ssDNA overhangs > ssDNA > dsDNA with 3′ ssDNA overhangs but not dsDNA with blunt ends and significantly that these interactions are increased after IR, suggesting that SAMHD1 can bind to multiple types of DNA substrates that can form at DSBs and that SAMHD1's binding to DNA is damage-regulated. Why SAMHD1 has preferential binding for 5′ ssDNA compared with 3′ ssDNA is not clear. As 3′ ssDNA overhangs must be generated at the DSB to facilitate HR, it is possible that there may be a greater requirement for SAMHD1 to bind to 5′ overhangs to then facilitate CtIP binding. In this regard, the endonuclease activity of pCtIP-MRN has a preference for targeting the 5′ terminated strand[5] and a number of studies have indicated that the 5′ terminated strand is preferentially degraded[101]. Recent studies using chemical crosslinking of ssRNA and

ssDNA oligonucleotides with SAMHD1 and x-ray crystallography of synthetic phosphorothioated oligonucleotides with SAMHD1 suggest that K354 is located in a region critical for SAMHD1's interface with nucleic acids[59,62]. Using a chemically synthesized peptide of FLAG-SAMHD1 (332–384), which was acetylated or not acetylated at K354, we found that K354 acetylation directly impairs SAMHD1 binding with ssDNA and furthermore that that SAMHD1 K354R has increased interactions with ssDNA and that K354Q has impaired interactions with ssDNA. Thus, our data support a model whereby acetylation at K354 leads to loss of electrostatic interactions with the negative charge of nucleic acid structures at DSBs. Although K354 also resides around the second allosteric binding site of dGTP[50,51,53], we found no evidence that K354 acetylation directs SAMHD1 tetramerization or dNTPase activity, suggesting that the acetylation status of K354 does not sufficiently contribute to dGTP binding. Our finding that SAMHD1 K354R but not K354Q rescues the impairment in IR-induced increase in CtIP-ssDNA binding of SAMHD1 depletion implies that SAMHD1's damage-regulated binding with ssDNA following K354 deacetylation in turn facilitates CtIP-ssDNA binding and combined with our previous observation that SAMHD1 facilitates CtIP recruitment to DSBs[69], provides further mechanistic insight into how SAMHD1 promotes DNA end resection by regulating CtIP's binding with ssDNA at DSBs.

SIRT1 has previously been shown to promote HR, at least in part, through deacetylation of BRG1[85], and possibly WRN[86,87]. We now define a role for SIRT1 in promoting DNA end resection that initiates HR through deacetylation of SAMHD1 at K354. Our finding that SAMHD1 K354R but not K354Q can fully rescue the impairment in DNA end resection of SIRT1 depletion but only partially rescue the HR impairment of SIRT1 depletion suggests that SIRT1 promotes DNA end resection primarily through SAMHD1 K354 deacetylation but promotes HR through deacetylation of multiple substrates, including SAMHD1 and other substrates such as BRG1 and WRN. The identification of SAMHD1 as a binding partner and substrate of SIRT1 adds to a growing number of SIRT1 substrates that function in promoting genome integrity, providing support for SIRT1 in regulating a network of proteins involved in the DDR. Moreover, our finding that SIRT1 directs DNA end resection through SAMHD1 deacetylation provides an additional layer of insight into how SIRT1 dysregulation leads to genomic instability and carcinogenesis. Finally, given that SAMHD1 depletion and proteasomal degradation sensitizes cancer cells to IR, PARP inhibitor, and other DSB-inducing agents[69], our findings provide mechanistic rationale for targeting the end resection function of SAMHD1 through SIRT1 inhibition as an adjunct to DSB-inducing agents for cancer therapy.

## Methods
### Cell Lines
HEK293T, HCT116, HeLa, and U2OS mammalian cell lines were purchased from American Type Culture Collection (ATCC, Manassas, VA). U2OS-235 mCherry-LacI-Fok1 cell line were provided by Dr. Roger Greenberg[95] and U2OS-DR-GFP cell line were obtained from Dr. Jeremy Stark[93]. AsiSI-ER-U2OS (DIvA-DSB inducible via AsiSI) cells were provided by Dr. Gaëlle Legube[97]. All cell lines were grown in DMEM (Gibco)

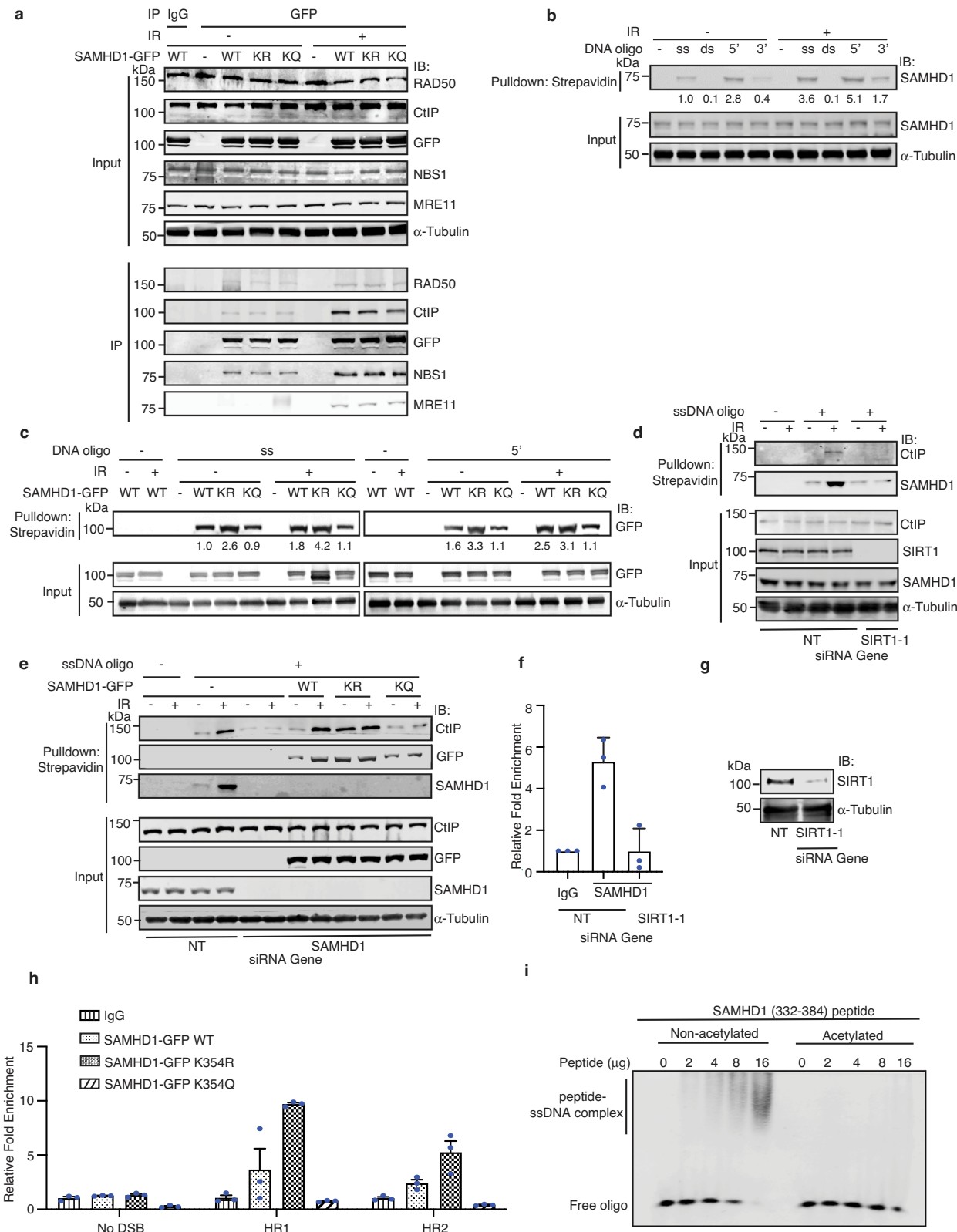

supplemented with 10% FBS. U2OS-235 was additionally cultured in 2 μg/mL puromycin and 100 μg/mL hygromycin. DIvA cells were additionally cultured in 1 μg/mL puromycin.

### Cell and protein lysate treatments

Cells were treated with 10 Gy of IR for 4 h using X-Ray 320 irradiator (Precision X-Ray Inc., N. Brandford, CT) and 50 J/m2 of UV for 3 h using

Spectrolinker XL-1500 UV crosslinker (Spectronics Corp., Westbury, NY). The following drug treatments were used to treat cells: 5 μM Camptothecin (CPT; Sigma # C9911) for 4 h; 0.5 μM Trichostatin A (TSA; Sigma #T8552) for 6 h; 10 mM Nicotinamide (NAM; Sigma # N3376) for 6 h; 1 μM Ex-527 (Sigma # E7034) for 6 h; 20 ug/mL Methyl methanesulfonate (MMS; Sigma # 129925) for 1 hour; 30 μM BrdU (BD Biosciences # 347580) for 40 h; 1 μM Shield-1 (Takara # 632189) for 4 h;

**Fig. 6 | SAMHD1 K354 deacetylation promotes binding with ssDNA and ssDNA overhangs and AsiSI-induced DSBs. a** 293T cells with or without SAMHD1-GFP and with or without IR were IP'ed with anti-GFP antibody to determine interaction of SAMHD1 with CtIP and the MRN complex. Input and IP'ed lysates probed with indicated antibodies are shown. **b** 293T cell lysates with and without IR were incubated with biotin-labeled DNA oligos and pulled down with agarose-streptavidin beads. Pull downs were separated by SDS-PAGE and immunoblotted with indicated antibodies. DNA oligos used were single-strand (ss), double-strand (ds), 5′ overhang (5′) and 3′ overhang (3′). **c** 293T cells overexpressing SAMHD-GFP were treated with IR followed by streptavidin-biotin pull-down as described in **b** with ss and 5′ overhang DNA. Anti-GFP and α-Tubulin antibodies were used for immunoblotting. **d** 293T protein lysates with or without SirT1 and with and without IR were incubated with biotinylated ssDNA and subjected to pull-down as described in **b**. **e** 293T silenced for SAMHD1 or a NT control were transfected with SAMHD1-GFP. Protein lysate were incubated with biotin-ssDNA and subjected to agarose-streptavidin pull down and western analysis. **f** ChIP-qPCR analysis of enrichment of SAMHD1 at HR prone DNA repair sites in DIvA cells transfected with SIRT1 or NT siRNAi is shown. DSBs were induced by treating DIvA cells with 500 nM of 4-OHT for 4 h. Mean and SEM of technical triplicates from a representative experiment is shown. Two independent experiments were performed with consistent results. **g** Western blot showing SIRT1 knockdown in cells used in **f**. **h** ChIP-qPCR analysis of enrichment of SAMHD1-GFP proteins at no DSBs, HR1, and HR2 prone repair sites in DIvA cells is shown. Mean and SEM of three technical replicates is shown from one experiment. Two biological replicates were done with consistent results. **i** Increasing amounts of non-acetylated or acetylated K354 SAMHD1 peptide (described in Fig. S6g) were incubated with biotin-ssDNA oligo for the EMSA assay. Free ssDNA and ssDNA complexed with the peptides were detected by Streptavidin conjugated to IR Dye 800. Source data are provided as a Source Data file.

2 mM or 500 nM 4-Hydroxytamoxifen (4-OHT; Sima # H7904) for 4 h to induce Fok1- or AsiSI-dependent DSB, respectively. For TSA, NAM and Ex-527 treatments, cells were pre-treated with the drug for 2 h and then treated for an additional 4 h in the presence of IR. For BrdU treatment, cells were pre-treated for 36 h and then treated for an additional 4 h in the presence of IR. Protein lysates for biotin-DNA pull-down assays were treated with 10 µL Rnase A (Sigma # R6148) for 45 min at 4 °C and 45 min at room temperature or with 10 µL Rnase H (NEB # M0297S) for 1.5 h at 37 °C.

## Antibodies

Primary antibodies used for western blotting, immunoprecipitation (IP) and immunofluorescence (IF) are as follows: SAMHD1 (Origene # TA502024; 1:1000 for Western, 1 µL per 1 mg lysate for IP; and Abcam # ab177462; 1:5000 for Western, 1:100 for IF); IgG (Invitrogen # 10500 C and Sigma # N103); Pan acetylated lysine (Cell Signaling Technology # 9441 S; 1:1000 for Western, 1 µL per 1 mg lysate for IP); GFP (Santa Cruz Tech # SC996; 1:1000 for Western; and Abcam #ab290; 1:5000 for Western, 1 µg per 2 mg lysate for IP); SirT1 (Abcam # 32441; 1:5000 for Western, 2 µL per 1 mg lysate for IP); SirT2 (ThermoFisher, custom made; 1:1000 for Western); SirT6 (Abcam # ab62738; 1:2000 for Western); SirT7 (Abcam # 62748; 1:1000 for Western); FLAG (Cell Signaling Technology # 2368 S; 1:1000 for Western; and Santa Cruz Tech # sc51590; 1:1000 for Western); α-Tubulin (Sigma # T6074; 1:10,000 for Western); GAPDH (Sigma, G9545; 1:2000 for Western); SAMHD1 K354Ac (Pierce, custom made; 1:1000 for Western); γH2AX (Cell Signaling Technology # 9718; 1:200 for IF; and Millipore # 05-636; 1:4000 for IF, 1:1000 for Western); BrdU (BD Biosciences # 347580; 1:200 for IF); RPA32 (Santa Cruz Tech # sc-14692; 1:400 for Western); pRPA32 S4/8 (Bethyl # A700-009; 1:1000 for Western); CtIP (Millipore # MABE1060; 1:1000 for Western); RAD50 (Abcam #ab228935; 1:500 for Western); MRE11 (Abcam # ab30725; 1:1000 for Western); NBS1 (Abcam # ab23996; 1:2000 for Western); H2AX (Bethyl # A300-082A; 1:800 for Western); p-CHK2 (Cell Signaling #2661; 1:100 for IF); GST (Santa Cruz Tech # sc-138; 1:1000 for Western). Secondary antibodies used for Western (at 1:10, 000) are: donkey anti-rabbit IR Dye 800 (Licor Biosciences #926-32213); donkey anti-rabbit IR Dye 680 (Licor Biosciences # 926-68023); donkey anti-mouse IR Dye 800 (Licor Biosciences # 926-32213); donkey anti-mouse IR Dye 680 (Licor Biosciences # 926-68022); Streptavidin-conjugated IR Dye 800 (Licor Biosciences # 926-32230). Secondary antibodies used for IF (at 1:1000) are: goat anti-mouse Alexa Fluor 555 (Invitrogen # A21424); goat anti-rabbit Alexa Fluor 647 (Invitrogen # A22144); goat anti-rabbit Alexa Fluor 488 (Invitrogen # A11034).

## Plasmids and siRNA

Plasmids expressing full-length SAMHD1-GFP and SAMHD1-RFP were generated by cloning SAMHD1 into the BamH1 and EcoR1 restriction sites of pcDNA3.1-GFP (Addgene # 70219) and RFP (Addgene # 13032), respectively. These plasmids were used as templates to generate SAMHD1-GFP/RFP KR, KQ and HD/AA mutants with Quick-Change site-directed mutagenesis kit (Agilent). To generate SAMHD1-GFP truncation mutants used to map the SirT1-binding domain in SAMHD1 and in DSB Fok1 assay, pcDNA3.1-SAMHD1-GFP was also used to PCR SAMHD1 fragments using primers containing BamH1 and EcoR1 sites (the myc NLS sequence was inserted between the SAMHD1 sequence and restriction site at the 3′ reverse primer, where needed) and cloning the fragment into pcDNA3.1-GFP (done by Emory Integrated Genomics Core). PET14b-6xHis-SAMHD1 bacterial plasmid expressing full-length SAMHD1 with a 6x His-tag at the N-terminus was a gift from Dr. Baek Kim. The plasmid was used to purify SAMHD1 and as template for the construction of his-tagged SAMHD1 K354R and K354Q by QuickChange site-directed mutagenesis (Agilent). SAMHD1-HA was generated by cloning SAMHD1 into pKH3 (Addgene, # 12555) using EagI and XbaI restriction sites. pcDNA-3XFLAG plasmid was obtained from Emory Integrated Genomics Core and pECE-hSirT1-FLAG-WT and FLAG-HY plasmids expressing SirT1 were a gift from Dr. David Gius. pCBASceI expressing I-Sce endonuclease was obtained from Addgene. The following siRNAs were used: SAMHD1-1 (CAACCAGAGCUGCAGAUAA); SAMHD1-5′UTR (ACGCAUGCUGAAGCTAAGTAA); SIRT1-1 (Dharmacon # D-003540-05); SIRT1-2 (Dharmacon # D-003540-06); SIRT2 (Dharmacon # D-004826-05); SIRT6 (Dharmacon # D-013306-030 and SIRT7 (Dharmacon # L-007774-01).

## Transfections

For plasmid transfections, 1–4 million cells were seeded on 60 mm plates and 24 h post plating, cells were transfected with 2–5 µg of the indicated plasmid using Lipofectamine 3000 as per manufacturer's instructions (ThermoFisher). Transfection medium was exchanged to fresh medium and cells were moved to 10 cm plates 4–6 h post-transfection. Cells were harvested 2–4 days post-transfection for downstream assays.

## RNAi silencing

SAMHD1 and Sirtuins were silenced using RNAi Max reagent using the protocol outlined by the manufacturer (Invitrogen). In brief, 30–60 nmol siRNA was transfected into one million cells in 60 mm plates. Medium was exchanged to fresh medium 24 h post transfection and grown for additional 24–48 h as needed for subsequent assays.

## Protein lysate preparation and western blotting

Harvested cells were washed twice in PBS buffer and then resuspended in 0.75% CHAPS buffer (10% glycerol, 150 mM NaCl, 50 mM Tris pH 7.5, 1 mM DTT, 1 µM TSA, 20 µM NAM, 0.75% CHAPS, protease inhibitors). Cells were lysed for 30 min on ice, then an equal amount of 0% CHAPS buffer was added (to final of 0.375% CHAPS) and lysate was clarified by centrifugation. When required (for H2AX westerns), lysates were sonicated four times, each for 10 s with 1 min on ice. Protein

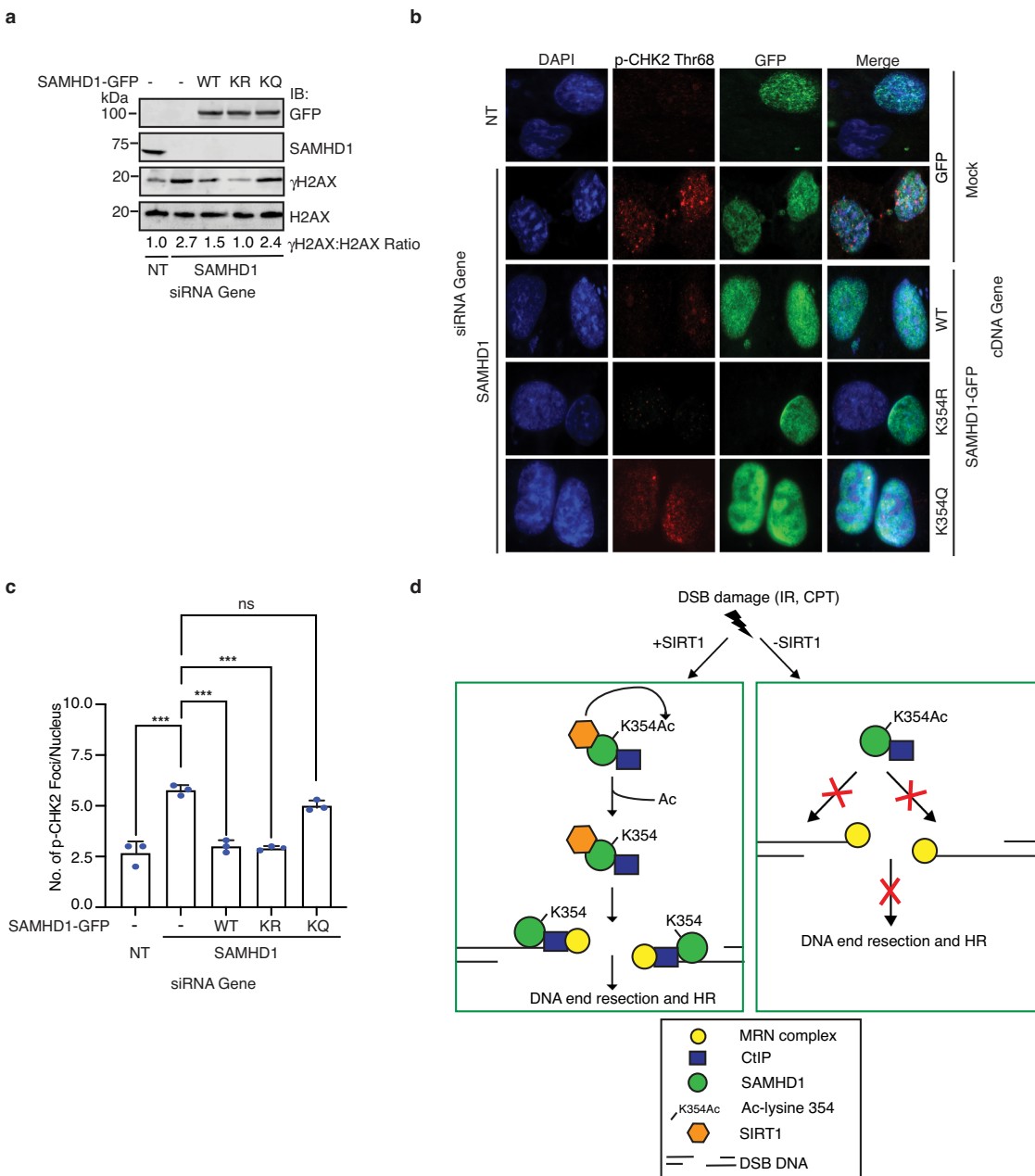

**Fig. 7 | SAMHD1 acetylation at K354 causes genomic instability. a** U2OS cells were transfected with SAMHD1-GFP WT, K354R, K354Q, or an empty vector and silenced with SAMHD1 or a NT siRNA control. Lysates were run on SDS-PAGE and probed with the indicated antibodies. **b, c** U2OS cells were transfected with siRNA against SAMHD1 or a NT siRNA control and also transfected with empty vector or vector expressing WT, K354R, or K354Q SAMHD1. Cells were subjected to immunofluorescence analysis with antibodies against p-CHK2 and GFP and DNA was stained with DAPI. The number of p-CHK2 foci per nucleus was counted. Shown are a representative image (**b**) and quantification of p-CHK2 foci represented by mean and SD of 50 cells from 3 independent experiments (**c**). *P* values (***p* < 0.001) were determined using Ordinary one-way ANOVA with Dunnett's post hoc test analysis. ns non-significant. **d** Model showing SAMDH1 deacetylation at K354 by SIRT1 in promoting DNA end resection and HR. In response to DSBs, SIRT1 complexes with and deacetylates SAMHD1 at K354, facilitating its recruitment and binding to ssDNA and ssDNA overhangs at DSBs. SAMHD1 binding to ssDNA at DSBs in turn facilitates CtIP binding to DSBs to promote DNA end resection and HR. Dysregulation of this pathway impairs DNA end resection and HR, leading to genomic instability. Source data are provided as a Source Data file.

concentration was determined by Bradford. Lysate was resolved by SDS-PAGE gel, transferred to PVDF and probed with the antibodies of interest. Where indicated, lysate was first used in IP experiments before running on gel. Signal was detected using the Li-Cor Odyssey system with the ImageStudio 5.2 software.

**Immunoprecipitation**

Protein lysate prepared as described above (500 μg–1 mg if IP is for an over-expressed protein and 2–4 mg if IP is for an endogenous protein) was precleared with 30 μL of protein G or protein A agarose beads (Roche # 10042392 and Invitrogen # 15918-014, respectively) for one hour. Precleared lysate was added to 30 μL protein G or A agarose beads that were prebound to the IP antibody or to 20 μL preconjugated Anti-Flag M2 affinity beads (Sigma # F2426) and incubated overnight on a rotator at 4 °C. Beads were washed three time with 0.375% CHAPS buffer on a rotator at room temperature and then resuspended in 50/50 0.375% CHAPS buffer and 5X SDS buffer. Resuspended beads were boiled for 8 min at 100 °C, centrifuged and supernatant was processed for Western blotting as indicated.

## DR-GFP assay

U2OS DR-GFP cells were transfected with the siRNA of interest. Twenty-four hours later, medium was removed, and cells were transfected with 3 µg I-SceI and, where required, 2 µg of either the empty RFP plasmid or the indicated SAMHD1-RFP rescue plasmid. Seventy-two hours later, cells were harvested, washed twice with PBS, resuspended in PBS, and subjected to flow cytometry (using SpectroFlo software 2.2.0.4 on the Aurora Cytek equipement) for GFP and, where indicated, RFP fluorescence. To measure HR efficiency, percentage of GFP positive cells, which represent HR positive cells, within the RFP positive cells (ie. when transfected with empty or rescue plasmids) was analyzed (Supplementary Fig. 3a, b) using the FlowJo software (version 10.8.1).

## BrdU foci DNA end resection assay

U2OS cells were transfected with the siRNA of interest (non-silencing or SAMHD1 siRNA) and twenty-four hours later, transfected with a SAMHD1-GFP plasmid, where indicated. Forty-eight hours after plasmid transfection, cells were incubated with 30 µM BrdU for 36 h and then incubated for an additional 4 h in the presence of IR. Cells were incubated in extraction buffer 1 (10 mM PIPES pH 7.0, 300 mM Sucrose, 100 mM NaCl, 3 mM MgCl$_2$, 1 mM EGTA, 0.5% Triton X-100) for 20 min on ice, washed with PBS and then incubated in extraction buffer 2 (10 mM Tris-HCl pH 7.5, 10 mM NaCl, 3 mM MgCl$_2$, 1% Tween 40, 0.5% sodium deoxycholate) for 15 min on ice. Cells were washed in PBS and subsequently fixed with PFA and permeabilized with Triton X-100 for immunofluorescence as described below with the appropriate antibody. DNA was visualized with DAPI (Southern Biotech # 0100-20) and SAMHD1-GFP was observed by GFP fluorescence.

## Immunofluorescence

U2OS cells and U2OS cells treated with SirT1 siRNA or Ex-527 were irradiated with 10 Gy IR for 4 h. U2OS-265 mCherry-LacI-Fok1 cells transfected with a SAMHD1-GFP plasmid, where indicated, was treated with Shield-1 and 4-OHT for 4 h to induce recombinant Fok1 expression. All cells were grown on coverslips prior to treatments. Coverslips were washed with PBS, fixed in 4% paraformaldehyde for 10 min at room temperature and permeabilized with 0.5% Triton X-100 for 10 min at 4 °C. Cells were then blocked for 15 min at room temperature in 5% BSA and probed with the indicated primary and secondary antibodies for 1 h each at 37 °C. DNA was stained with DAPI and coverslips were mounted on slides. Immunofluorescence signals were visualized by Leica SP8 inverted confocal microscope at ×63 magnification using the LasX (3.5.2.18963) software.

## Protein purification and analytical size exclusion chromatography

pET14b-6xHis bacterial plasmids expressing SAMHD1 proteins were expressed in *E. coli* BL21 Rosetta cells, induced by 0.3 mM isopropyl-1-thio-D-galactopyranoside (IPTG) at 16 °C, and further grown overnight. After cells lysis by sonication on ice, proteins were purified by His-Trap affinity chromatography (GE Healthcare). Protein was further purified by Superdex 200 size exclusion chromatography (SEC) (GE Healthcare) in a buffer containing 20 mM Tris pH 7.8, 50 mM NaCl, 1 mM DTT, and 5% glycerol. For SEC, SAMHD1 proteins (5 µM) were preincubated with dGTP (3 mM) or without dGTP, and the mixtures (200 µL) were applied to an analytical Superdex 200 column (10 × 300 mm) at a flow rate of 0.5 mL/min. The column was calibrated with gel filtration standard (BioRad) and equilibrated with a buffer containing 20 mM Tris-HCl, pH 7.8, 50 mM NaCl, 10 µM dGTP, 5 mM MgCl2, and 5% glycerol. The elution profiles with UV traces of excitation at 280 nm were recorded.

## Cellular dNTP pool quantification

293 T cells not expressing or expressing SAMHD1-GFP WT, K354R, K354Q, or HD/AA were washed in cold PBS and then resuspended in 65% methanol for lysis. Cell lysate was heated to 95 °C, clarified by centrifugation, dried in speed vac and resuspended in water. Quantification of cellular dNTP pool was carried out as outlined previously[31]. Briefly, $^{32}$P-5′ labeled primer (18 nucleotide) was annealed to four different 19-nucleotide templates creating nucleotide variation at the template 5′ end. dNTPs from cell extracts were used to extend a single nucleotide at the 5′ end of the primer. Products were ran on urea-PAGE gels and amount of extended primers for each of the 4 variations was quantified with BioRad PharosFX imager and Image Lab software (version 5.12). Amount dATP, dGTP, dTTP, and dCTP in cells were expressed as pmol/$10^6$ cells.

## Biotinylated DNA pull-down assay

293T or 293T cells expressing WT, K354R, or K354Q SAMHD1-GFP were not irradiated or irradiated with 10 Gy IR for 4 h. Cells were washed with PBS and cell lysates were prepared by resuspending in Buffer B (10 mM Tris-HCl pH 7.5, 100 mM NaCl, 10% Glycerol, 10ug/mL BSA, 0.05% NP40, CHAPS 0.35%, protease inhibitors) and incubating on ice for 20 min. Lysates were clarified by centrifugation and protein concentration in the supernatant was determined by Bradford assay. Whole cell lysates (600 µg–1 mg) were precleared with 30 µL streptavidin-conjugated agarose beads (Millipore # 69203) for 1 h at 4 °C, where the beads had been pre-washed in Buffer A (10 mM Tris-HCl pH 7.5, 100 mM NaCl). Precleared lysates were incubated with 30 µL streptavidin-conjugated beads pre-bound to 40 pmol biotinylated DNA overnight at 4 °C. As a negative control, precleared lysate was also incubated with beads not prebound to biotinylated DNA. Biotinylated DNA used were single-stranded (ss), double-stranded (ds), 5′ overhang or 3′ overhang and sequences are as previously described (Fig. 5a). Lysate-beads-biotinylated samples were washed with Buffer B four times, boiled in 5X SDS sample buffer and western blotted with SAMHD1 or GFP antibody.

## In-vitro GST pulldown

Recombinant GST (Abcam #ab89494) or GST-SIRT1 (containing amino acids 193-741 of SIRT1) (Active Motif # 31533) were pre-incubated with glutathione-agarose beads (Pierce # 16100) in equilibration buffer (50 mM Tris-Cl pH 8, 150 mM NaCl, protease inhibitors) for one hour at 4 °C with rotation. No SAMHD1 or recombinant SAMHD1 was added and the mixture was further incubated for 2 h at 4 °C on the rotator. The beads were washed with equilibration buffer a few times, resuspended in SDS sample buffer, boiled and subjected to SDS-PAGE electrophoresis and western blotting with antibodies against SAMHD1 or GST. Two micrograms of the proteins used in the GST pulldown assays were also subjected to SDS-PAGE electrophoresis, followed by Coomassie-staining of the gel.

## Clonogenic assay

U2OS, U2OS silenced for endogenous SAMHD1 or U2OS silenced for endogenous SAMHD1 and expressing SAMHD1-GFP WT, K354R or K354Q proteins were plated at 500 cells per well in six-well plates 48 h post transfection. Cells were then treated with Ex-527 (1 µM) and indicated concentrations of veliparib and allowed to proliferate for 10–12 days. Cells were subsequently fixed with methanol and stained with crystal violet reagent. Surviving colonies containing greater than 50 cells were counted and plotted from three independent experiments.

## ChIP assay

ChIP assay was performed using ChromaFlash™ One-Step ChIP Kit (P-2025, Epigentek, Farmingdale, NY, USA) according to the manufacture's instruction. Briefly, DIvA cells were fixed with 1% formaldehyde (Sigma-252549) for 15 mins at room temperature. Next, 0.125 M glycine was added for 5 min to stop excessive crosslinking reaction of the formaldehyde. Cells were then washed thrice with ice-

cold PBS for 5 mins and lysate was prepared using 0.375% CHAPS in lysis buffer (10% glycerol, 150 nM NaCl, 50 nM Tris pH-7.5). The lysed cells were sonicated to an average size of 500–800 base pairs using a Branson microtip SFX250 sonicator. Anti-GFP or anti-SAMHD1 was used for immunoprecipitation for 4 h at room temperature with IgG as the negative control. The immunoprecipitated chromatin was subjected to DNA isolation and quantitative real-time (PowerTrack™ SYBR Green Master Mix A46012, PCR 7500 Fast Real-Time PCR system, ThermoFisher) was performed using the following primers specific for no-DSBs, HR1 and HR2 prone sites in DIvA cells:

no-DSBs: Forward: 5′-ATTGGGTATCTGCGTCTAGTGAGG-3′
Reverse: 5′- GACTCAATTACATCCCTGCAGCT-3′
HR1: Forward: 5′-GATTGGCTATGGGTGTGGAC-3′
Reverse: 5′- CATCCTTGCAAACCAGTCCT-3′
HR2: Forward: 5′-CCGCCAGAAAGTTTCCTAGA-3′
Reverse: 5′- CTCACCCTTGCAGCACTTG-3′

### Electrophoretic mobility shift assay
Varying amounts of chemically synthesized FLAG-tagged non-acetylated or acetylated K354 SAMHD1 peptides (DYKDDDDKKRFIKFARV-CEVDNELRICARDKEVGNLYDMFHTRNSLHRRAYQHKVGNIIDT or DYKDDDDKKRFIKFARVCEVDNELRICARD-Lys(Ac)-EVGN-LYDMFHTRNSLHRRAYQHKVGNIIDT, respectively; synthesized by LifeTein) were incubated with 0.2 μM of biotin-labeled ssDNA oligo (Fig. S6a, b) in 1X binding buffer (25 mM Tris-Cl, 50 mM KCl, 4.2 mM $MgCl_2$, 5% glycerol, 1 mM EDTA, 250 μM dNTP, pH 7.9) for 30 min at room temperature. Samples containing ssDNA oligo without peptides were also included as a negative control. Samples were mixed with 1X DNA loading buffer, ran on non-denaturing acrylamide gel and transferred to nylon membrane. Unbound DNA and DNA-protein complexes were crosslinked to the nylon membrane at 120 mJ/cm² for 45 s using the Spectrolinker XL-1500 UV crosslinker (Spectronics Corp., Westbury, NY) and blotted for biotin-ssDNA detection with Streptavidin conjugated to IR Dye 800. Signal was detected using Li-Cor Odyssey system.

### Mass spectrometry
SAMHD1-HA was IP'ed from 293T cells (not-treated or treated with CPT or IR) with anti-HA conjugated agarose beads (monoclonal, mouse; Sigma A2095; 1 μg per 2 mg protein lysate) as described above. 293T cells not expressing SAMHD1-HA was also IP'ed as a negative control. IP'ed material from 4 samples were processed for mass spectrometry to identify SAMHD1 acetylation sites. To prepare samples for mass spectrometry, digestion buffer (200ul of 50 mM $NH_4HCO_3$) was added to the IP'ed beads and samples were treated with 1 mM dithiothreitol (DTT) at 25 °C for 30 min, followed by 5 mM iodoacetimide (IAA) at 25 °C for 30 min in the dark. Protein was digested with 1:50 (w/w) lysyl endopeptidase (Wako) at 25 °C overnight. Samples were then diluted with 50 mM $NH_4HCO_3$ to a urea concentration below 2 M and further digested overnight with 1:50 (w/w) trypsin (Promega) at 25 °C. Resulting peptides were desalted with a Sep-Pak C18 column (Waters) and dried under vacuum. For LC-MS/MS analysis, derived peptides were resuspended in 10 μL of loading buffer (0.1% formic acid, 0.03% trifluoroacetic acid, 1% acetonitrile). Peptide mixtures (2 μL) were separated on a self-packed C18 (1.9 μm Dr. Maisch, Germany) fused silica column (25 cm × 75 μM internal diameter (ID); New Objective, Woburn, MA) by a Dionex Ultimate 3000 RSLCNano and monitored on a Orbitrap Fusion mass spectrometer (ThermoFisher Scientific, San Jose, CA). Elution was performed over a 1200 min gradient at a rate of 275 nl/min with buffer B (0.1% formic in acetonitrile) ranging from 3% to 40%. The mass spectrometer cycle was programmed to collect at the top speed for 3 s cycles. The MS scans (400–1600 m/z range, 200,000 AGC, 50 ms maximum ion time) were collected at a resolution of 120,000 at m/z 200 in profile mode and the HCD MS/MS spectra (1.5 m/z isolation width with 0.5 m/z offset, 30%

collision enegy, 10,000 AGC target, 35 ms maximum ion time) were detected in the ion trap. Dynamic exclusion was set to exclude previous sequenced precursor ions for 20 s within a 10 ppm window. Precursor ions with +1, and +8 or higher charge states were excluded from sequencing. For data processing, spectra were searched using Proteome Discoverer 2.0 against human Uniprot database (90,300 target sequences). Searching parameters included fully tryptic restriction and a parent ion mass tolerance (± 20 ppm). Methionine oxidation (+15.99492 Da), ubiquitination (+114.04293 Da), phosphorylation (+79.96633 Da), lysine acetylation (+42.01057 Da), and protein N-terminal acetylation (+42.01057 Da) were variable modifications (up to 3 allowed per peptide); cysteine was assigned a fixed carbamido-methyl modification (+57.021465 Da). Percolator was used to filter the peptide spectrum matches to an false discovery rate of 1%.

### Statistics and reproducibility
Experiments quantified and represented by graphs, along with the representative western blots and/or immunofluorescence images corresponding to them, were repeated 2–5 times, as indicated in the figure legends. Each data point is shown (blue circle). Graphs were made using GraphPad Prism 9. All other western blots and immunofluorescence studies were repeated at least three times. SAMHD1 tetramerization experiments (Fig. 4a, b) were repeated twice. EMSA (Fig. 7i) experiments were repeated five times. Mass spectrometry study (Fig. S2) was conducted once. All experiments repeated independently gave similar results.

### Reporting summary
Further information on research design is available in the Nature Portfolio Reporting Summary linked to this article.

## Data availability
The data that support this study are available from the corresponding author upon request. The mass spectrometry proteomics data have been deposited to the ProteomeXchange Consortium via the PRIDE partner repository with the dataset identifier PXD031357. Source data are provided with this paper.

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

## Acknowledgements

We thank Dr. Roger Greenberg for U20S-235 mCherry-LacI-Fok1 cells, Dr. Jeremy Stark for U20S-DR-GFP cells, Dr. Gaelle Legube for AsiSI-ER-U2OS cells, and Dr. David Gius for PECE-SIRT1-FLAG WT and H363Y plasmids. We thank members of the Yu lab for their helpful discussion and technical expertize. We thank Dr. Pritha Bagchi (Emory Integrated Proteomics Core) for helpful discussion with mass spectrometry data. This work was supported by: National Institutes of Health (NIH) National Cancer Institute (NCI) (R01CA178999 and R01CA254403 to D.S.Y.); Department of Defense Breast Cancer Research Program (BC180883 to D.S.Y.); NIH (R01 CA255257 to X.D.); NIH (AI136581 and AI162633 to B.K.) and NIH (F31 AI157884 to N.E.B.). Research reported in this publication was supported in part by the Emory Integrated Genomics Core (EIGC) Shared Resource of Winship Cancer Institute of Emory University and NIH/NCI under award number P30CA138292. The content is solely the responsibility of the authors and does not necessarily represent the official views of the NIH.

## Author contributions

P.K. and D.S.Y. conceived and designed the study. P.K. S.R., X.L, Z.S., N.E.B, Y.C., R.H. W.D., E.V.M, D.D., D.F., and D.M.D. performed and analyzed the experiments. P.K. S.R., X.L, Z.S., N.E.B, Y.C., R.H., W.D., E.V.M, D.D., D.F., D.M.D, N.T.S., X.D., E.A.O., B.K., and D.S.Y. interpreted the findings. P.K. and D.S.Y. wrote the manuscript with input from all authors.

## Competing interests

The authors declare no competing interests.
