## [Peer Review File · Nature Communications]

SAMHD1 deacetylation by SIRT1 promotes DNA end resection by facilitating DNA binding at double-strand breaksREVIEWER COMMENTS

Reviewer #1 (Remarks to the Author):

In this paper, Kapoor-Vazirani et al. reported that SAMHD1 is deacetylated by SIRT1 on K354 which facilitates its binding with ssDNA at DSBs. This process affects DNA end resection and HR through facilitating CtIP ssDNA binding to maintain genome integrity. Figures 1-5 are mostly solid, and the data support the authors' conclusion. I have some problems with Figure 6 and the physiological significance of this regulation.

In figure 2, I was surprised that the ac-K354 antibody can recognize the KQ mutant. This has been shown for other Ac-specific antibodies?

In Figure 3g, why does overexpression of SAMHD1 WT in SIRT1 knockdown cells have a similar effect as K354R? There is no SIRT1 to deacetylate SAMHD1, WT should be acetylated? This also brings up the issue of % SAMHD1 that is acetylated vs deacetylated before and after DNA damage.

In Figure 6a, when they want to show the acetylation status does not affect the interaction between SAMHD1 and CtIP and other proteins, mock-IR conditions should also be used.

How does SIRT1 or SAMHD1 affect CtIP foci at DSBs?

The authors claimed that SAMHD1 K354 deacetylation promotes genome stability. However, they only showed the rH2AX level by western. More evidence is needed.

Not clear why SAMHD1 binds preferentially to DNA with 5' overhang, how does this fit into the current model of end resection, as CtIP works at the initiation of end resection to introduce a nick to generate 3' overhang.

Based on the authors' conclusion, the status of SAMHD1 K354 acetylation affects HR. Then the SAMHD1 mutant K354R and K354Q should show different sensitivity to PARPi. The authors should examine the sensitivity to PARPi in cells expressing WT/K354R/K354Q SAMHD1 at least in vitro, and if possible in vivo. It's also been reported that several PARPi niraparib, olaparib, rucaparib, talazoparib, veliparib, PJ34, and XAV939 do not affect SIRT1 activity. If one combines the SIRT1 inhibitor Ex527 with the PARPi treatment, would the cells become more sensitive?

Reviewer #2 (Remarks to the Author):

In this manuscript, the authors show that SIRT1-mediated deacetylation of SAMHD1 promotes its ssDNA-binding activity to stimulate DNA end resection and HR at double-strand breaks (DSBs). Mutations in SAMHD1 are associated with an inherited autoimmune encephalopathic disorder, Aicardi-Goutières syndrome. SAMHD1 has been reported to play a role in maintenance of genomic stability. They show that SIRT1 deacetylates SAMHD1 at K354 after DNA breakage. The deacetylation of SAMHD1 promotes ssDNA-binding activity and the recruitment of CtIP, and thus stimulates DNA resection and HR repair of DSBs.

This is an interesting, well-conducted, and clearly presented study, which provides insights into the acetylation-mediated regulation of SAMHD1 in maintenance of genomic stability. However, the mechanism of how acetylation of SAMHD1 regulates its ssDNA-binding is not clear (see below).

Major comments:

1. Depletion of SAMHD1 strongly reduced HR (Figure 3). Likely, SAMHD1 is required for DNA end

resection and HR as important as CtIP (previous results by the same group in Cell Reports 2017). CtIP is essential gene for cells, while mutations in BRCA1, which is less important for HR than CtIP, induce cancer in human and mouse. However, why Aicardi-Goutières syndrome deficient in SAMHD1 does not show cancer predisposition and SAMHD1-deficient mice is almost normal.

2. Figure 6, acetylation of SAMHD1 reduces its ssDNA-binding. All in vitro assays in this figure used cell extracts. Thus it's hard to judge if acetylation of SAMHD1 directly effects its ssDNA-binding activity or indirectly through its interactions with other factors. Purified wild-type or mutant SAMHD1 is necessary to be used for examining this point in EMSA assay.

3. Figure 5, Does SAMHD1 300-465 region have ssDNA-binding activity in vitro EMSA assay? Does acylation at K345 site effect it?

4. Why SAMHD1 tends to bind 5' ssDNA overhangs compared to 3' ssDNA overhang? Is it related to the physiological event?

Minor comments:

1. Figure 1c, the panels of "IP" and "Input" should cut from the same film, but not separated. Otherwise, it's hard to judge if SIRT2, 6 and 7 are not detected indeed in the "IP" when using the same expose time.

2. Figure 2b, the quality of Tubulin band after Ac-Lys IP is low.

3. Figure 6a, the quality of this figure is low.

4. Because the differences in Figure 6c, d and k are low, quantifying the results will be better.

Reviewer #3 (Remarks to the Author):

In the manuscript titled "SAMHD1 deacetylation by SIRT1 promotes DNA end resection by facilitating DNA binding at double-strand breaks", the authors investigate the regulation of SAMHD1 by SIRT1-mediated deacetylation at K354 in response to DNA double-strand breaks. Deacetylated SAMHD1 is recruited to the site of DNA damage. It also promotes DNA end resection via ssDNA binding and facilitates CtIP ssDNA binding and HR repair.

Overall, this manuscript describes a series of carefully executed experiments with proper controls. Additionally, it shows new mechanistic insights into how SIRT1 and SAMHD1 in regulating DNA DSB repair. This study is potentially important to the DNA repair field. However, there are a few concerns that need to be addressed to support the conclusion.

1. The deacetylation is IR dependent, while the SIRT1-SAMHD1 binding is also IR dependent. Do they bind directly? And How exactly do SIRT1 and SAMHD1 interact? A domain mapping experiment is crucial.

2. An epistatic study using double depletion of SIRT1 and SAMHD1 in the DR-GFP reporter assay will provide a better understanding of whether SAMHD1 functions solely or partially downstream of SIRT1

3. What is the possible explanation for the nuclear localization for the mutants lacking NLS?

4. This study should also provide in vitro evidence that SAMHD1 can bind to the DNA substrates directly.

5. It is not entirely clear to me, at least in this study, how SAMHD1 is regulated in the context of DSB recruitment. Genetic studies will help delineate the role of SAMHD1 in the HR pathway.

6. There is a discrepancy with the previous report (Daddacha W, 2017) from the same group on SAMHD1 IRIF formation. In the current study, SAMHD1 doesn't seem capable to form foci (Figure 5). The subcellular localization of full length seems inconsistent. Fig 5A shows nuclear while Fig 5E shows cytoplasmic localization. Can the authors clarify?

Also, the current manuscript showed subcellular localization difference between K354R and K354Q. Does endogenous SAMHD1 translocate after DNA damage (beside foci) over time?

7. How exactly does SAMHD1 physically interact with the MRN complex?

8. A survival assay is needed to support the role of K354 acetylation in DNA repair besides gammaH2AX marker.

9. What's the mechanistic prediction or explanation on the acetylated SAMDH1 in ssDNA affinity?

10. The author stated that SAMHD1-GFP expressed in 293T can pull down nuclear sirtuins. It is unclear how the samples were prepared to obtain "nuclear", but not the sirtuins potentially in non-nuclear localization.

September 11, 2022

Manuscript Number: NCOMMS-22-01909

"SAMHD1 deacetylation by SIRT1 promotes DNA end resection by facilitating DNA binding at double-strand breaks"

Dear Reviewers,

We greatly appreciate the thorough and thoughtful review of our manuscript entitled "SAMHD1 deacetylation by SIRT1 promotes DNA end resection by facilitating DNA binding at double-strand breaks" for consideration for publication as an Article in *Nature Communications*. The additional results and control experiments uniformly support our original interpretation of the data, but they make a much stronger argument, thanks to your feedback and suggestions. In this regard, the following changes and additional data have been added to the revised manuscript to address the suggestions and concerns. Specifically, 5 new results panels have been added to the revised manuscript: (1) Fig. 3f; 3g (2) Fig. 6i; (3) Fig. 7b, 7c; and 11 new results panels have been added to the Supplementary section: (1) Fig. S1b, S1c; (2) Fig. S3b; (3) Fig. S4; (4) Fig. S5c; (5) Fig. S6e, S6g, S6h; (6) Fig. S7a, S7b, S7c. In addition, we have replaced a number of results panels with additional controls at the request of the Reviewers: (1) Fig. 1c; (2) Fig. 2c, 2e; (3) Fig. 5a; (4) Fig. 6a, Fig. 6b, Fig. 6c (5) Fig. 7a. Thus, the new manuscript now has 7 figures containing a total of 50 panels, and the Supplementary section now has 7 figures containing a total of 23 panels for a total of 73 figure panels in this Article.

The revised manuscript addresses the comments of the Reviewers and includes new data that uniformly supports our previous conclusions, strengthens our original interpretation of the data, and provides further insight into the mechanism by which SAMHD1 deacetylation by SIRT1 promotes DNA end resection by facilitating DNA binding at double-strand breaks (DSB). Significantly, we now present evidence that a chemically synthesized peptide of FLAG-SAMHD1 (332 – 384) directly binds to ssDNA using both electrophoretic mobility shift assay (EMSA) and streptavidin pulldown of biotinylated ssDNA, and that this binding is impaired when the peptide is acetylated at K354, suggesting that K354 acetylation directly impairs SAMHD1 binding to ssDNA. We also present evidence that bacterially purified recombinant SIRT1 and SAMHD1

interact *in vitro*, suggesting that their interaction is direct; SAMHD1-GFP (300-465) but not SAMHD1-GFP (1-115) or (115-300) interacts with SIRT1; and SAMHD1 (300-465) shows increasing binding to ssDNA in response to IR, consistent with K354 as a SIRT1 deacetylation site and SAMHD1-GFP (300-465) but not SAMHD1-GFP (1-115) or (115-300) localizing to DSBs. We also show through an epistasis study that SIRT1 and SAMHD1 function together in promoting HR, and moreover, that SAMHD1 depletion in cells causes hypersensitivity to veliparib, a PARP inhibitor, which is rescued by expression of SAMHD1-GFP WT and K354R but not K354Q, providing further support that SAMHD1 deacetylation at K354 promotes HR. In addition, we found that SIRT1 depletion, similar to SAMHD1, impairs CtIP localization to DSBs. We also present evidence that SAMHD1 depletion in cells leads to an increase in p-CHK2 Thr56 foci, which is rescued by expression of SAMHD1-GFP WT and K354R but not K354Q, providing further support for our original interpretation of the data that SAMHD1 K354 deacetylation promotes genome stability. Finally, we provide improved immunofluorescence, co-IP, and western blot images, a number of which we experimentally reproduced, quantification of streptavidin pulldown and genomic instability studies, and additional discussion and clarification text as requested by the Reviewers.

In summary, we believe that the additional revisions have significantly improved the overall scientific quality of this study. The individual comments and our detailed responses follow. In each case, the responses are non-italicized, and the appropriate language that has been added to the manuscript text is in blue.

We hope the revisions to the manuscript are acceptable to you.

Respectfully,

David Yu, M.D., Ph.D.
Associate Professor and Jerome Landry, MD, Professor of Cancer Research
Director of the Division of Cancer Biology
Department of Radiation Oncology
Emory University School of Medicine
dsyu@emory.edu

Reviewer #1, General Comments:

In this paper, Kapoor-Vazirani et al. reported that SAMHD1 is deacetylated by SIRT1 on K354 which facilitates its binding with ssDNA at DSBs. This process affects DNA end resection and HR through facilitating CtIP ssDNA binding to maintain genome integrity. Figures 1-5 are mostly solid, and the data support the authors' conclusion. I have some problems with Figure 6 and the physiological significance of this regulation.

We greatly appreciate Reviewer #1 in taking the time to review our manuscript and provide the thoughtful comments and suggestions. Please see comments below.

Reviewer #1, Specific Comments:

Major Points:

1. In figure 2, I was surprised that the ac-K354 antibody can recognize the KQ mutant. This has been shown for other Ac-specific antibodies?

We agree with the Reviewer that recognition of the KQ mutant, albeit less than for WT, by the site-specific anti-acetyl SAMHD1 K354 antibody is somewhat surprising. In retrospect, we should not have included this data in the original submission as it is not a commonly observed phenotype as astutely pointed out by the Reviewer and have since revised this data from new Fig. 2e. Given that this antibody recognizes endogenous SAMHD1 and exogenous SAMHD1-GFP WT but not K354R expressed in 293T cells where the bands are also recognized by an anti-SAMHD1 antibody and can be silenced with SAMHD1 siRNA (new Fig. 2d-e), and furthermore our data using this anti-acetyl SAMHD1 K354 antibody show that endogenous SAMHD1 is deacetylated at K354 in response to IR, and this is rescued by SIRT1 depletion (new Fig. 2f), which is consistent with our mutational analyses indicating that SAMHD1 is predominantly acetylated at K354 and that K354 is the major site of deacetylation in response to IR (new Fig. 2b-c), we remain confident in the specificity of this antibody towards SAMHD1 K354.

2. In Figure 3g, why does overexpression of SAMHD1 WT in SIRT1 knockdown cells have a similar effect as K354R? There is no SIRT1 to deacetylate SAMHD1, WT should be acetylated? This also brings up the issue of % SAMHD1 that is acetylated vs deacetylated before and after DNA damage.

This is an interesting point. It is likely that when SAMHD1-RFP WT is overexpressed, there is a sufficient amount of non-acetylated SAMHD1-RFP WT that can alleviate the impairment in HR from SIRT1 deficiency similar to overexpression of SAMHD1-RFP K354R (new Fig. 3i). This may be because there is insufficient endogenous acetyltransferase to acetylate all of the overexpressed SAMHD1-RFP WT. We have seen a similar rescue of HR impairment with overexpression of RFP-BARD1 WT and KR compared with KQ in SIRT2 deficient cells (Minten et al, Cell Reports, 2021). Our data indicate that under endogenous conditions in multiple cell types, including 293T, HeLa, HCT116, and U2OS cells, SAMHD1 is acetylated under nondamaged conditions (new Fig. 1a), deacetylated specifically in response to DSB inducing agents IR and CPT but not UV or MMS (new Fig. 1a and Supplementary Fig. S1a), and that this acetylation does not

significantly increase under nondamaged conditions with treatment of cells with trichostatin A (TSA), which inhibits Class I, II and IV HDACs, or nicotinanide (NAM), which inhibits Class III HDACs (sirtuins) (new Fig. 1b), suggesting that SAMHD1 does not have basal deacetylation in the absence of DSB inducing agents. Because trypsin cleaves at lysine, it is often difficult to get the modified and unmodified peptide to calculate an exact percentage of acetylation at a given residue by mass spectrometry; however, our data in new Fig. 1a, Fig. 2f, and Supplementary Fig. S1a suggest that most of acetylated SAMHD1, including at K354, is deacetylated in response to DSB-inducing agents.

3. In Figure 6a, when they want to show the acetylation status does not affect the interaction between SAMHD1 and CtIP and other proteins, mock-IR conditions should also be used.

As well suggested, we examined the interaction of SAMHD1 WT, K354R, and K354Q with CtIP and the MRN complex in cells both before and after IR and observed a comparable damage-induced increase in pulldown of CtIP and the MRN complex with SAMHD1 WT, K354R, and K354Q (new Fig. 6a), supporting our original interpretation of the data that K354 deacetylation is not critical for mediating interaction with CtIP or the MRN complex.

4. How does SIRT1 or SAMHD1 affect CtIP foci at DSBs?

We performed the well suggested experiment and found that depletion of SIRT1 or SAMHD1 in U2OS cells impairs CtIP localization to IR-induced foci that co-localize with γ H2AX (new Supplementary Fig. S7). These data are consistent with our prior report showing that SAMHD1 promotes CtIP recruitment to DNA damage sites (Daddacha et al., Cell Reports, 2017).

5. The authors claimed that SAMHD1 K354 deacetylation promotes genome stability. However, they only showed the γ H2AX level by western. More evidence is needed.

We agree with the Reviewer in regards to this point. As such, we examined for the spontaneous accumulation of CHK2 phosphorylation at Thr-56, a marker for genomic instability, by immunofluorescence microscopy, and found that SAMHD1 depletion in U2OS cells leads to an increase in p-CHK2 Thr56 foci, which is rescued by expression of SAMHD1-GFP WT and K354R but not K354Q (new Fig. 7b-c). Together with our original data showing that SAMHD1 depletion in U2OS cells causes an increase in γ H2AX but not total H2AX levels, which can be rescued by expression of SAMHD1-GFP WT and K354R but not K354Q (new Fig. 7a), these data strongly support our original interpretation of the data that SAMHD1 K354 deacetylation promotes genome stability.

6. Not clear why SAMHD1 binds preferentially to DNA with 5' overhang, how does this fit into the current model of end resection, as CtIP works at the initiation of end resection to introduce a nick to generate 3' overhang.

This is an interesting observation, which we have further elaborated on in the Discussion section. As 3' ssDNA overhangs must be generated at the DSB to facilitate HR, it is possible that there may be a greater requirement for SAMHD1 to bind to 5' overhangs to then facilitate CtIP binding. In this regard, the endonuclease activity of pCtIP-MRN has a preference for targeting the 5'

terminated strand (Anand et al., Mol Cell, 2016) and a number of studies have indicated that the 5' terminated strand is preferentially degraded (Sun et al., Cell, 1991; White et al., EMBO J, 1990; Zhou et al., NAR, 2014; Zakharyevich et al., Mol Cell, 2010). Dissecting why SAMHD1 has preferential binding to 5' ssDNA to direct CtIP activity is an area of future investigation in our lab although outside of the scope of the current study focused on the upstream signaling events governing SAMHD1 function in DNA end resection.

7. Based on the authors' conclusion, the status of SAMHD1 K354 acetylation affects HR. Then the SAMHD1 mutant K354R and K354Q should show different sensitivity to PARPi. The authors should examine the sensitivity to PARPi in cells expressing WT/K354R/K354Q SAMHD1 at least in vitro, and if possible in vivo. It's also been reported that several PARPi niraparib, olaparib, rucaparib, talazoparib, veliparib, PJ34, and XAV939 do not affect SIRT1 activity. If one combines the SIRT1 inhibitor Ex527 with the PARPi treatment, would the cells become more sensitive?

These are all excellent points. We performed the well suggested experiment and found that SAMHD1 depletion in cells causes hypersensitivity to veliparib, a PARP inhibitor, which is rescued by expression of SAMHD1-GFP WT and K354R but not K354Q (new Supplementary Fig. S4), providing further support for our original interpretation of the data that SAMHD1 deacetylation at K354 promotes HR.

We also performed the well suggested experiment of combining the SIRT1 inhibitor Ex-527 with PARP inhibitor treatment but found no evidence that SIRT1 inhibition with Ex-527 sensitizes U2OS cells to veliparib or rucaparib (see figure below), which may be because both SIRT1 and PARP have diverse substrates that could impact cell survival, have an interplay through a common NAD⁺ cofactor, PARP1 is negatively regulated by SIRT1 (Rajamohan et al., MCB, 2009), and as astutely pointed out by the Reviewer, SIRT1 activity is not regulated by PARP inhibition (Ekblad et al., ChemBioDrugScreen 2015). Since negative data in this type of experiment is difficult to interpret unambiguously as it does not prove that SIRT1 inhibition does not sensitize cells to PARP inhibitor, we have not included this data in the manuscript.

Reviewer #2, General Comments:

In this manuscript, the authors show that SIRT1-mediated deacetylation of SAMHD1 promotes its ssDNA-binding activity to stimulate DNA end resection and HR at double-strand breaks (DSBs). Mutations in SAMHD1 are associated with an inherited autoimmune encephalopathic disorder, Aicardi-Goutières syndrome. SAMHD1 has been reported to play a role in maintenance of genomic stability. They show that SIRT1 deacetylates SAMHD1 at K354 after DNA breakage. The deacetylation of SAMHD1 promotes ssDNA-binding activity and the recruitment of CtIP, and thus stimulates DNA resection and HR repair of DSBs.

This is an interesting, well-conducted, and clearly presented study, which provides insights into the acetylation-mediated regulation of SAMHD1 in maintenance of genomic stability. However, the mechanism of how acetylation of SAMHD1 regulates its ssDNA-binding is not clear (see below).

We greatly appreciate Reviewer #2 for taking the time to review our manuscript and provide the thoughtful comments and suggestions.

Reviewer #2, Specific Comments:

Major Points:

1. Depletion of SAMHD1 strongly reduced HR (Figure 3). Likely, SAMHD1 is required for DNA end resection and HR as important as CtIP (previous results by the same group in Cell Reports 2017). CtIP is essential gene for cells, while mutations in BRCA1, which is less important for HR than CtIP, induce cancer in human and mouse. However, why Aicardi-Goutières syndrome deficient in SAMHD1 does not show cancer predisposition and SAMHD1-deficient mice is almost normal.

These are all important points. While SAMHD1-deficient Aicardi-Goutières Syndrome cells demonstrate genomic instability (Kretschmer et al., BMJ, 2005 and Franzolin et al., FASEB, 2019) and SAMHD1 has been reported to be recurrently mutated and/or dysregulated in a number of cancers, including leukemias, lymphomas, colorectal cancer, and lung cancer, as astutely pointed out by the Reviewer, there is thus far no evidence that Aicardi-Goutières Syndrome patients deficient in SAMHD1 show cancer predisposition or that SAMHD1 deficient mice develop cancer. It is possible that genetic background and/or environmental factors in the context of SAMHD1 deficiency may be necessary for tumorigenesis, which albeit outside the scope of this study, would be interesting to investigate.

2. Figure 6, acetylation of SAMHD1 reduces its ssDNA-binding. All in vitro assays in this figure used cell extracts. Thus it's hard to judge if acetylation of SAMHD1 directly effects its ssDNA-binding activity or indirectly through its interactions with other factors. Purified wild-type or mutant SAMHD1 is necessary to be used for examining this point in EMSA assay.

This is an excellent suggestion. To determine whether K354 deacetylation is critical for mediating a direct interaction with ssDNA, we performed an electrophoretic mobility shift assay (EMSA)

with a chemically synthesized peptide of FLAG-SAMHD1 (332 – 384), which was acetylated or not acetylated at K354 (new Supplementary Fig. S6g). A dose dependent shift in mobility of ssDNA was observed with an increasing amount of non-acetylated SAMHD1 (332 – 384) peptide but not appreciably with K354 acetylated SAMHD1 (332 – 384) (new Fig. 6i), suggesting that SAMHD1 (332 – 384) is sufficient for binding ssDNA and that K354 acetylation directly impairs SAMHD1 binding to ssDNA. In a complementary approach, streptavidin pulldown of biotin labeled ssDNA confirmed the binding of non-acetylated SAMHD1 (332 – 384) peptide to ssDNA, which was impaired by K354 acetylation (new Supplementary Fig. S6h), implying that K354 deacetylation is critical for mediating a direct interaction with ssDNA.

3. Figure 5, Does SAMHD1 300-465 region have ssDNA-binding activity in vitro EMSA assay? Does acylation at K345 site effect it?

See above. In addition, we also performed streptavidin pulldown of biotin labeled ssDNA using SAMHD1-GFP (300-465) purified from cells treated with and without IR and found that SAMHD1 (300-465) showed increased binding with ssDNA in response to IR (new Supplementary Fig S6e). Our *in vitro* EMSA assay described above indicate that an even smaller domain of SAMHD1 (332-384) directly binds ssDNA, and that this binding is impaired with K354 acetylation.

4. Why SAMHD1 tends to bind 5' ssDNA overhangs compared to 3' ssDNA overhang? Is it related to the physiological event?

This is an interesting observation, which we have further elaborated on in the Discussion. As 3' ssDNA overhangs must be generated at the DSB to facilitate HR, it is possible that there may be a greater requirement for SAMHD1 to bind to 5' overhangs to then facilitate CtIP binding. In this regard, the endonuclease activity of pCtIP-MRN has a preference for targeting the 5' terminated strand (Anand et al., Mol Cell, 2016) and a number of studies have indicated that the 5' terminated strand is preferentially degraded (Sun et al., Cell, 1991; White et al., EMBO J, 1990; Zhou et al., NAR, 2014; Zakharyevich et al., Mol Cell, 2010). Dissecting why SAMHD1 has preferential binding to 5' ssDNA to direct CtIP activity is an area of future investigation in our lab although outside of the scope of the current study focused on the upstream signaling events governing SAMHD1 function in DNA end resection.

Minor Points:

1. Figure 1c, the panels of "IP" and "Input" should cut from the same film, but not separated. Otherwise, it's hard to judge if SIRT2, 6 and 7 are not detected indeed in the "IP" when using the same expose time.

We thank the Reviewer for bringing this important point to our attention. As requested, we experimentally reproduced new Fig. 1c showing the IPs and inputs all on the same blot and found that co-IP of SAMHD1-GFP in cells pulls down endogenous SIRT1 but not endogenous SIRT2, 6, or 7 following IR treatment, supporting our original interpretation of the data that SAMHD1 complexes with endogenous SIRT1 but not SIRT2, 6, or 7 in a damage regulated manner.

2. *Figure 2b, the quality of Tubulin band after Ac-Lys IP is low.*

We agree with the Reviewer in regards to this point. As such, we experimentally reproduced the IP of Ac-lys with an improved tubulin control IP blot (new Fig. 2c), and found that SAMHD1 K354R and to a lesser extent K405R are acetylated at lower levels, suggesting that both K354 and K405 contribute to SAMHD1 acetylation. Furthermore, IR induced a decrease in acetylation in all of the mutants except K354R. Our findings are consistent with our data obtained with IP of SAMHD1-GFP (new Fig. 2b) and support our original interpretation of the data that K354 is the major site of deacetylation in response to IR.

3. *Figure 6a, the quality of this figure is low.*

We agree with the Reviewer and have now experimentally reproduced new Fig. 6a, which shows that co-IP of SAMHD1-GFP WT, KR, and KQ in cells pulls down comparable amounts of endogenous CtIP, MRE11, RAD50, and NBS1 following IR treatment, supporting our original interpretation of the data that K354 deacetylation is not critical for mediating interaction with CtIP or the MRN complex.

4. *Because the differences in Figure 6c, d and k are low, quantifying the results will be better.*

As requested, we quantified the results in new Fig. 6b, 6c, and new Fig. 7a

Reviewer #3, General Comments:

In the manuscript titled “SAMHD1 deacetylation by SIRT1 promotes DNA end resection by facilitating DNA binding at double-strand breaks”, the authors investigate the regulation of SAMHD1 by SIRT1-mediated deacetylation at K354 in response to DNA double-strand breaks. Deacetylated SAMHD1 is recruited to the site of DNA damage. It also promotes DNA end resection via ssDNA binding and facilitates CtIP ssDNA binding and HR repair. Overall, this manuscript describes a series of carefully executed experiments with proper controls. Additionally, it shows new mechanistic insights into how SIRT1 and SAMHD1 in regulating DNA DSB repair. This study is potentially important to the DNA repair field. However, there are a few concerns that need to be addressed to support the conclusion.

We greatly thank the reviewer for taking the time to review our manuscript and provide the thoughtful comments and suggestions.

1. The deacetylation is IR dependent, while the SIRT1-SAMHD1 binding is also IR dependent. Do they bind directly? And How exactly do SIRT1 and SAMHD1 interact? A domain mapping experiment is crucial.

These are all excellent suggestions. As requested, we examined the *in vitro* interaction of bacterially recombinant SAMHD1 and bacterially recombinant GST-SIRT1 and found that GST-

SIRT1 but not GST alone pulls down SAMHD1 (new Supplementary Fig. S1b-c), suggesting that the interaction of SAMHD1 and SIRT1 is direct.

As well suggested, we mapped the interaction of SAMHD1-GFP fragments with SIRT1-FLAG expressed in cells and found that SAMHD1-GFP (300-465) but not SAMHD1-GFP (1-115) or (115-300) co-IP's with SIRT1-FLAG in irradiated cells (new Supplementary Fig. S5c). These data are consistent with K354 as a SIRT1 deacetylation site, SAMHD1-GFP (300-465) but not SAMHD1-GFP (1-115) or (115-300) localizing to DSBs (new Fig. 5e-f), and SAMHD1-GFP (300-465) showing increased binding to ssDNA in response to IR (new Supplementary Fig. S6e). Collectively, these data indicate that SAMHD1-GFP (300-465) is sufficient for interaction with SIRT1, localization to DSBs, and binding to ssDNA.

2. An epistatic study using double depletion of SIRT1 and SAMHD1 in the DR-GFP reporter assay will provide a better understanding of whether SAMHD1 functions solely or partially downstream of SIRT1

We performed the well-suggested experiment and found that both SIRT1 and SAMHD1 depletion alone in cells impairs HR, and combined depletion of SIRT1 and SAMHD1 causes no further impairment in HR (new Fig. 3f-g and Supplementary Fig. S3b), suggesting that SIRT1 and SAMHD1 function together in promoting HR.

3. What is the possible explanation for the nuclear localization for the mutants lacking NLS?

We should have been clearer in describing this data. We have clarified in the text of the Results and Figure Legends sections that “Mutants lacking SAMHD1’s nuclear localization signal (NLS), KRPR at amino acids 11-14, were fused to myc-NLS to ensure nuclear localization.”

4. This study should also provide in vitro evidence that SAMHD1 can bind to the DNA substrates directly.

This is an excellent suggestion. To determine whether SAMHD1 K354 deacetylation is critical for mediating a direct interaction with ssDNA, we performed an electrophoretic mobility shift assay (EMSA) with a chemically synthesized peptide of FLAG-SAMHD1 (332 – 384), which was acetylated or not acetylated at K354 (new Supplementary Fig. S6g). A dose dependent shift in mobility of ssDNA was observed with an increasing amount of non-acetylated SAMHD1 (332 – 384) peptide but not appreciably with K354 acetylated SAMHD1 (332 – 384) (new Fig. 6i), suggesting that SAMHD1 (332 – 384) is sufficient for binding ssDNA and that K354 acetylation directly impairs SAMHD1 binding to ssDNA. In a complementary approach, streptavidin pulldown of biotin labeled ssDNA confirmed the binding of non-acetylated SAMHD1 (332 – 384) peptide to ssDNA, which was impaired by K354 acetylation (new Supplementary Fig. S6h), implying that K354 deacetylation is critical for mediating a direct interaction with ssDNA.

5. It is not entirely clear to me, at least in this study, how SAMHD1 is regulated in the context of DSB recruitment. Genetic studies will help delineate the role of SAMHD1 in the HR pathway.

Our data indicate that SIRT1 depletion or inhibition with Ex-527 in cells impairs the co-localization of endogenous SAMHD1 with IR-induced DSBs marked by γ H2AX (new Fig. 5a and Supplementary Fig. S5a) and that SIRT1 depletion or inhibition impairs the co-localization of SAMHD1 with mCherry-LacI-FokI endonuclease-induced DSBs in U2OS reporter cells integrated with lac operator repeats (new Fig. 5b and Supplementary Fig. S5b), suggesting that SIRT1 deacetylase activity promotes SAMHD1 localization to DSBs. Combined with our data showing that SAMHD1-GFP K354R but not K354Q co-localizes with FokI-induced DSBs (new Fig. 5e-f), collectively, these data strongly suggest that SIRT1 deacetylation of K354 facilitates SAMHD1 recruitment to DSBs.

6. There is a discrepancy with the previous report (Daddacha W, 2017) from the same group on SAMHD1 IRIF formation. In the current study, SAMHD1 doesn't seem capable to form foci (Figure 5). The subcellular localization of full length seems inconsistent. Fig 5A shows nuclear while Fig 5E shows cytoplasmic localization. Can the authors clarify? Also, the current manuscript showed subcellular localization difference between K354R and K354Q. Does endogenous SAMHD1 translocate after DNA damage (beside foci) over time?

We should have been clearer in presenting this data. We have now provided improved quality images that more clearly show endogenous SAMHD1 co-localizing with IR-induced DSBs marked by γ H2AX (new Fig. 5a), consistent with those published in Daddacha et al., Cell Reports, 2017, and demonstrating reproducibility of our findings.

Differences in subcellular localization of SAMHD1 WT as well as for K354R and K354Q observed in new Fig. 5a vs 5e are likely attributed to the overexpression of SAMHD1-GFP in Fig. 5e compared with endogenous SAMHD1 in Fig. 5a as well as non-nucleoplasmic fluorescence, that is more clearly observed when SAMHD1 localizes to FokI-induced DSBs.

In new Fig. 5b, we also show that endogenous SAMHD1 localizes to FokI-induced DSBs. We have also previously shown that endogenous SAMHD1 localizes to DNA damage sites induced by laser microirradiation (Supplementary Fig. S2 in Daddacha et al., Cell Reports, 2017).

7. How exactly does SAMHD1 physically interact with the MRN complex?

Using microscale thermophoresis with recombinant SAMHD1 and MRE11, the Pasero group showed that SAMHD1 directly binds to MRE11 (Coquel, et al., Nature, 2018). Since we observed comparable pulldown of endogenous MRE11, RAD50, and NBS1 when we co-IP'd SAMHD1-GFP WT, K354R, and K354Q expressed in 293T cells (new Fig. 6a), suggesting that SAMHD1 K354 deacetylation is not critical for mediating interaction with the MRN complex, we did not further characterize the interaction of SAMHD1 with the MRN complex.

8. A survival assay is needed to support the role of K354 acetylation in DNA repair besides gammaH2AX marker.

We agree with the Reviewer in regards to this point and found that SAMHD1 depletion in cells causes hypersensitivity to veliparib, a PARP inhibitor, which was rescued by expression of SAMHD1-GFP WT and K354R but not K354Q (new Supplementary Fig. S4). Given that HR

impairment is associated with PARP inhibitor hypersensitivity, these data provide further support for our original interpretation of the data that SAMHD1 deacetylation at K354 promotes HR.

9. What's the mechanistic prediction or explanation on the acetylated SAMHD1 in ssDNA affinity?

This is an excellent point which we have further elaborated on in the Discussion section. Recent studies using chemical crosslinking of ssRNA and ssDNA oligonucleotides with SAMHD1 and x-ray crystallography of synthetic phosphorothioated oligonucleotides with SAMHD1 suggest that K354 is located in a region critical for SAMHD1's interface with nucleic acids. Using a chemically synthesized peptide of FLAG-SAMHD1 (332 – 384), which was acetylated or not acetylated at K354, we found that K354 acetylation directly impairs SAMHD1 binding with ssDNA (new Fig. 6i and Supplementary Fig. S6g-h) and furthermore that SAMHD1 K354R has increased interactions with ssDNA and that K354Q has impaired interactions with ssDNA (new Fig. 6c and e). Thus, our data support a model whereby acetylation at K354 leads to loss of electrostatic interactions with the negative charge of nucleic acid structures at DSBs.

10. The author stated that SAMHD1-GFP expressed in 293T can pull down nuclear sirtuins. It is unclear how the samples were prepared to obtain "nuclear", but not the sirtuins potentially in non-nuclear localization.

We should have been clearer in describing this data. We have clarified in the text of the Results section that we “analyzed for pull down of the non-mitochondrial sirtuins SIRT1, SIRT2, SIRT6 and SIRT7 that localize to the nucleus.” Samples prepared were from whole cell extracts as described in the Methods section.

REVIEWERS' COMMENTS

Reviewer #1 (Remarks to the Author):

This manuscript revealed a novel regulation of SAMHD1 by SIRT1 in DNA repair. I think the authors have satisfactorily addressed my concerns. I support the publication of this manuscript.

Reviewer #2 (Remarks to the Author):

The authors have satisfactorily answered all my concerns and I would recommend the manuscript for publication in Nature Communications.

Reviewer #3 (Remarks to the Author):

In the revised manuscript, the authors have significantly improved the manuscript by including new supporting data and addressing all the questions with detailed clarification. The experiments were well-designed and executed. I think the discovery will be high impact and important to the genome instability field. I support this high-quality work to be published.

October 11, 2022

The individual comments from the Reviewers and our detailed responses also follow. In each case, the responses are non-italicized.

Reviewer #1 (Remarks to the Author):

This manuscript revealed a novel regulation of SAMHD1 by SIRT1 in DNA repair. I think the authors have satisfactorily addressed my concerns. I support the publication of this manuscript.

Thank you.

Reviewer #2 (Remarks to the Author):

The authors have satisfactorily answered all my concerns and I would recommend the manuscript for publication in Nature Communications.

Thank you.

Reviewer #3 (Remarks to the Author):

In the revised manuscript, the authors have significantly improved the manuscript by including new supporting data and addressing all the questions with detailed clarification. The experiments were well-designed and executed. I think the discovery will be high impact and important to the genome instability field. I support this high-quality work to be published.

Thank you.